# A vignette study of option refusal and decision deferral as two forms of decision avoidance: Situational and personal predictors

**Sabrina Berens**[ORCID]**, Joachim Funke***

Department of Psychology, Heidelberg University, Heidelberg, Germany

* Joachim.Funke@psychologie.uni-heidelberg.de

## Abstract

The avoidance of a decision is a phenomenon that has been studied in various forms and psychological disciplines. Nevertheless, previous studies often lacked the integration of situational as well as personal factors in predicting decision avoidance. Additionally, studies about conditions that affect different forms of decision avoidance are still lacking. Therefore, this study investigated how situational and personal factors influenced two different forms of decision avoidance: 1) the option of deferring choice to a later point in time (*decision deferral*) and 2) the option to refuse both alternatives (*option refusal*). Furthermore, this study aimed to analyze how the participants experienced their avoidance behavior. This served the purpose of capturing functional and dysfunctional avoidance out of a subjective perspective instead of providing predefined best options. A vignette study based on realistic decision-making scenarios within a student's daily life was conducted. In an online survey, N = 312 participants chose to decide, to defer choice, or to refuse options in the context of eight vignettes. Situational factors (time pressure, lack of information, and attractiveness of alternatives) were systematically varied. Additionally, the following personal factors were captured by questionnaires: indecisiveness, decision-making styles, and the need for cognitive closure. Further factors were captured, but not systematically varied: selection difficulty, importance of a decision, and similarity of alternatives. The individual satisfaction with decision-making behavior was evaluated as an indicator of subjective decision quality. The results showed that decision deferral was affected by situational factors (primarily due to time pressure), selection difficulty, and individual indecisiveness. The refusal of current alternatives was exclusively influenced by situational factors, mainly the manipulated attractiveness of the choice-set. The results emphasize the functionality of refusing unattractive options and help to distinguish adaptive and maladaptive coping forms in decision-making.

## Introduction

Decision-making has mainly been explored concerning determinants of the act of choice. Thereby, the process character of a decision is missed, and determinants of decision avoidance

**Data Availability Statement:** All relevant data are within the manuscript and its Supporting Information files.

**Funding:** We acknowledge financial support by Deutsche Forschungsgemeinschaft (DFG) within

the funding program "Open Access Publishing" by the Baden-Württemberg Ministry of Science, Research and the Arts and by Ruprecht-Karls-Universität Heidelberg.

**Competing interests:** The authors have declared that no competing interests exist.

are neglected. Decision avoidance is a phenomenon that has been studied in various forms and psychological disciplines [1]: choice deferral of consumer decisions in marketing research (e.g., [2]), heuristic and biases of decision avoidance in cognitive psychology (e.g., [3]), decision inertia in critical incidents in naturalistic decision-making research (e.g., [4]), procrastination research in educational or business psychology (e.g., [5]), chronic indecisiveness in clinical psychology or personality psychology (e.g., [6]). Altogether, there is a bunch of studies in the context of decision avoidance, but they have explored the phenomenon out of the perspective of different and partly unlinked psychological disciplines. In line with Anderson [1], the present study aimed to complement different research traditions to a more comprehensive view. Therefore, this study wants to combine situational and personal predictors of decision avoidance and analyzes how participants experience their avoidance behavior.

## Forms of decision avoidance and a process model of action

Anderson [1] describes four different forms of decision avoidance: People often like to stay with the familiar (status quo bias [7]), they prefer omissions over actions in risky decision situations (omission bias [8]), omit action when a similar more attractive option was recently missed (inaction inertia [9]) or defer decisions to look for further options or information (choice deferral). The different forms of decision avoidance need not to reflect an inactive and passive process, but could be distinguished by avoidant intention vs. seeking intention in an active or passive way [10]: In the case of an avoidant intention, it could be a deliberate, but passive choice to do nothing (omission), but also a deliberate choice to take an action that results in no change (status quo). In the case of a seeking intention, there could be a difficulty in committing to a choice that results in a delay (procrastination), but it could also be an active and deliberate process to defer a decision (choice deferral). Studies on status quo bias and omission bias formulate hypothetical situation in which the decision-maker can do something (e.g., vaccinate or investing stocks) or omit from doing so and staying with the status quo, especially in a high-risk situation with potentially harmful consequences [7,8]. In there, a balancing of potential risks and the fear of negative consequences play an important role in the decision process. These are other basic conditions as in studies on choice deferral in consumer decisions. In there, hypothetical situations are formulated in which the decision-maker can decide on various products (e.g., buy camera A or B) or has the opportunity to search for further options. Thus, a balancing of preferences goes along with the option not to choose at all. These decisional situations should be less stressful unless specific situational circumstances increase the pressure (e.g., time pressure), uncertainty (e.g., lack of information) or discomfort (e.g., unattractive options) or if personal characteristics impair confident and rational decision making (e.g., chronic indecisiveness, anxiety).

Indecisiveness is a personality variable described as having chronic difficulties with decisions and deferring them as a result [11]. In this definition, the subjective experience of difficulty and decision-making behavior are linked. Indecisiveness is associated with psychopathological disorders [12]. In clinical psychology, indecisiveness is often considered as a form of anxiety, avoidance or lack of motivation or self-confidence. Studies within educational psychology do often classify choice deferral as a subdomain of procrastination. Procrastination is conceptualized as a personal disposition to postpone, delay, and thereby avoid actions or decisions that have to be taken [13]. In general, procrastination is characterized as a maladaptive pattern [14] that contradicts the personal aims by definition [15,16]. As a differentiation, decisional procrastination has been linked to high neuroticism and low self-regulation, while task avoidance procrastination is linked to low conscientiousness and self-control [13].

This study will focus on what Anderson [1] has described as choice deferral to mainly focus on the avoidance of choice as distinct from the avoidance of action. Previous studies often blur terms of decision avoidance by not separating between action avoidance and decision avoidance. The Rubicon Model of Action Phases [17] helps to classify forms of decision avoidance into a larger process model of action. In this model, the act of choice is the step from the predecisional to the preactional phase (= crossing the Rubicon), but essential motivational and volitional processes take place before and after [18]. Decision avoidance is part of the predecisional or deliberative phase, while the implementation of a decision within action is part of the postdecisional or implemental phase [19]. By taking a closer look at the different phases, it is possible to distinguish between decision avoidance and action avoidance: For example, one can decide whether to take Course A or Course B in the gym (predecisional phase). One can delay this decision for a longer time, e.g., because one is still thinking about what to like better (decision deferral). One can also simply decide against both options and do not take any course (option refusal). Then the decision process is either completed, or one looks for new alternatives (predecisional phase starts again). Assuming a decision for a course has been made, and one has registered (crossing the Rubicon). Then the decision process is completed, and the preactional phase begins. In there, one can postpone going to the decided course, e.g., because it is difficult to motivate oneself (action avoidance). In this study, the focus is on the predecisional phase, especially on the determinants of decision avoidance.

In real-life, decisions are often deferred to a later point in time to collect more information [20] or options are refused, because no alternative satisfies [21]. Anderson defines choice deferral as a form of decision avoidance that is characterized by either postponing a decision or refusing to select an option [1]. Previous studies in consumer research often combined both options in a no-choice condition (not chose either of these and look for others) [2,22,23]. Studies about conditions that affect different form of decision avoidance are still lacking, even though Anderson [1] had already pointed out this research gap 17 years ago. Therefore, we want to address this gap by analyzing how situational and personal factors influence two different forms of decision avoidance (decision deferral and option refusal). Hence, in our study, we separate these two forms of decision avoidance, so that a decision-maker has the possibility to either choose, defer choice (decision deferral), or refuse to choose any option (option refusal). Decision deferral is defined as to postpone the decision to a later point in time, while option refusal is a decision against existing options.

## Predictors of decisional avoidance

Heckhausen and Heckhausen [24] found (decisional) motivation to be influenced by personal as well as situational factors. As motivation is crucial for the decision of an avoidance option, it can be expected that the latter is influenced by personal and situational factors as well. The combination of personal indecision and the risk of a decision has already been studied [25,26]. Nevertheless, studies integrating situational and personal factors in predicting the choice of an avoidance option are rare. Therefore, this study aimed to consider situational and personal factors, which influence decision deferral and option refusal as two forms of decision avoidance. Context conditions that determine decisions were selected based on previous literature and relevance in everyday life. Personal characteristics were selected based on relevance for decision-making and avoidance behavior. Our study aimed to differentiate previous findings for the two different forms of decision avoidance. Hypotheses were formulated on two-staged manipulated variables in situational factors and on continuous variables for personal factors.

**Situational factors.**   Studies within social psychology consider choice deferral to be a general human tendency [1] and investigate conditions of decision avoidance. In this study,

different vignettes are designed to explore the impact of situational conditions on decision deferral and option refusal by varying them systematically. In there, decision situations were developed to represent common and meaningful situations in a student's everyday life. Previous studies primarily focused on the context of product purchases [2,22,23,27,28], but also other daily decision situations like Friday night activity [22], seminar [25,26] or apartment [23,29,30] have been studied. This study attaches importance to the fact that vignettes reflect familiar decisional situations that are formulated in a realistic way and fit the sample of student participants. Therefore, this study uses no ecological approach (like in field studies) but realistically simulated environments (like e.g., [31]) to combine concerns of naturalistic decision-making as well as controlled experimental scenarios [32]. Overall, many situational factors have been shown to influence decision avoidance, but no previous study has compared the different effects on decision deferral and option refusal as two different forms of decision avoidance.

An important decision framework is the time available for a decision. It is not clear whether time pressure reduces [23] or increases [33,34] selection difficulty. However, higher time pressure should reduce decision deferral, because the costs of deferral (e.g., risk that previous options change or that one loses options) could exceed possible benefits (e.g., the gain of information or clarity) the closer a deadline gets. Under a condition of low time pressure, decision deferral could be reasonable to search for further information or alternatives or to increase deliberation about consequences and to capture the complexity [13]. Besides the expected gain of information, deferral has its added value by increasing certainty with decision and acceptance of unpleasant decisions [35,36]. The relation between the length of time and the deferral and avoidance behavior has already been shown [23,30,36]. Our study seeks to replicate these findings in further, complementing decision situations and tests for the first time whether the effects differ between decision deferral and option refusal. In there, larger effects are expected on decision deferral than on option refusal, because in the context of high time pressure decision deferral implies no gain, but option refusal could still be reasonable.

**H1**: *Low time pressure implies more decision avoidance (especially decision deferral) than high time pressure.*

The completeness of the information mainly characterizes uncertainty in real-life decision situations [37]. A lack of information is likely to increase uncertainty and thus selection difficulty, since the options cannot be conclusively compared, and therefore it could remain open which option is superior. Decision avoidance could aim to reduce the lack of transparency by gathering further information [20,38,39]. This could be reasonable if the available time is sufficient [13]. Avoiding decisions under uncertainty has been investigated within lottery tasks, like the Ellsberg Paradox [40]. As these tasks are far from reality, the association between uncertainty and decision avoidance should be investigated with more realistic decision tasks, as done in this study. Furthermore, this study firstly investigates whether the effects differ between decision deferral and option refusal. Due to the higher uncertainty, which can be concluded with more time, decision deferral seems more evident than option refusal under the condition of a high lack of information. Nevertheless, it is possible that if both options are associated with an as yet unclear risk, they are more likely to be rejected.

**H2**: *Lack of information implies more decision avoidance (especially decision deferral) than full information.*

The nature of alternatives also influences decision-making. In there, quantity [41], distinctiveness [22,30], and attractiveness [23,27] of alternatives are considered relevant in studying decision avoidance. Decisions between two unattractive options (aversion-aversion conflict) are usually more difficult and more unpleasant than decisions between attractive options (appetence-appetence conflict [42,43]). Therefore, in decision situations of high stress, a

decision between two unattractive options is often avoided (as studied in status quo bias and omission bias [7,8]). Additionally, research about decisions in the context of critical incidents showed that redundant deliberation about negative consequences results in behavioral inaction and even worse outcomes (decision inertia, [4]). On the other hand, it seems reasonable to continue searching for new alternatives if time pressure is low, and previous options are unsatisfying [2,21]. Therefore, in the context of consumer decisions, option refusal increases if alternatives are not attractive [27]. In there, the avoidance seems like an adequate strategy and no dysfunctional avoidance. Therefore, this study investigates how the attractiveness of the choice-set influences the choice of decision deferral or option refusal in the context of daily decision situations. In there, larger effects are expected on option refusal than on decision deferral.

**H3**: *Decisions within an unattractive choice-set imply more decision avoidance (especially option refusal) than decisions within an attractive option-set.*

**Personal factors.** Additionally, decision avoidance can be considered from the perspective of a differential and cross-situational form of dealing with decision situations [44]. Important psychological constructs in this field are *decision-making styles* [45,46], personality variables like chronic *indecisiveness* [11], and the *need for cognitive closure* [47]. Other more general and not decision specific factors, such as anxiety and neuroticism, are taken into account in decision-making styles.

Decision-making styles describe the emotional, cognitive, and behavioral reactions within the decision-making process [48]. It is assumed that every person provides all styles, but there are individual patterns to react in stressful decision situations [44]. Decision-making styles influence the process of decision-making. So far, most of them have barely been studied in the context of decision avoidance. Nevertheless, ideas can be developed of how rational, intuitive, confident, anxious, and dependent decision-making styles influence decision avoidance according to their definition. In literature, different classifications of decision-making styles are proposed:

Maybe the most classical distinction is the one between rational and intuitive decision-making. Rational decision-makers weigh alternatives logically and with more attention to detail, whereas intuitive decision-makers act faster and more emotion-based [49]. The rational style requires more time to make decisions compared to the intuitive style [49]. Therefore, rational decision making could be associated with decision deferral because of a further search for information or deliberation. However, this association is also questionable because the rational style is often classified as adaptive and is contrasted to avoiding choice [48]. On the other hand, the intuitive style has its strength in automatic and fast decision-making [50], which is why this style is associated with a decreased deferral. Another important concept in this context is the confidence of the decision-maker [44], which describes a positive self-perception of oneself as a decision-maker [6]. In contrast, anxious decision-making describes the tendency of doubting one's own decisions, fearing negative consequences, and worrying a lot. The anxious decision style includes clinically relevant conditions of depression or anxiety disorders [6]. Confidence in decision-making should decrease avoidance behavior (especially decision deferral), while anxiety should increase avoidance behavior according to the conflict theory [44]. Social aspects can furthermore influence decision-making. In this context, dependent decision-making means that the decision-maker relies on the advice of others in the decision-making process [49]. A dependent decision-making style could increase avoidance behavior because it is associated with lower confidence [51] and asking for help also requires time, which justifies deferral. According to defensive avoidance, two styles pushed through: 1) buck-passing and 2) procrastination [48]. Buck-passing means to pass decisions to others and includes a wish to hand over responsibility, while procrastination describes the deferring of

choice. A deferral style is considered as a personality variable if it is used consistently and independently of situational risks [26]. Buck-passing implies decision avoidance [48] and should increase option referral, but not necessarily decision deferral. Because of the high intercorrelations of the different decision-making styles and the limited number of previous studies, they were only analyzed exploratory within correlational analyzes and not within the main analyses' models. The aim is to gain an impression of the association between different decision-making styles and the tendency to defer decisions and to refuse options.

Chronic indecisiveness is probably the most important personal factor identified in previous studies. Indecisive persons experience more difficulty and are less confident in their decision-making [52]. They have a higher threshold of feeling confident [53] and collect more information about the relevant alternatives [25]. Therefore, indecisiveness is expected to be a predictor of deferring choice [25,53,54]. Patalano and Wengrovitz [26] reported that indecisive participants did not show uniformly increased deferral, but rather that their deferral behavior is less strongly adapted to context conditions like increased risk. Therefore, this study wants to show if the indecisiveness of a person has an additional predictive effect compared to the situational factors in the prediction a decision avoidance behavior. Therefore, it is included as the main personal factor in the analysis models. It has never been studied before if chronic indecisiveness impacts decision deferral and options refusal as two options of decision avoidance differently. Since indecision is primarily associated with more time to decide, it is expected to be mainly associated with decision deferral and less with option refusal.

**H4**: *The higher the indecisiveness of a person, the more decision avoidance is shown (especially decision deferral).*

Another personality variable relevant to the investigation of decision avoidance is the need for cognitive closure (NCC). NCC is a construct that influences information processing [48] and the duration of decision-making processes [36,55]. A high NCC describes the tendency to accelerate decision-making processes by searching for less information or contradictions [47]. A high NCC as a differential variable can have advantages in case of an approaching deadline, but can have disadvantages if important aspects are neglected [55]. Referring to the Rubicon Model of Action Phases, the predecisional phase is characterized by broad information processing [18], for which a low NCC is considered to be crucial. On the other hand, the postdecisional phase already focuses on implementation, which requires focusing rather than to broaden the view, and therefore requires a high NCC [55]. It could be expected that the NCC influences the two forms of decision avoidance differently: A low NCC is associated with a broadening of the decision process and should, therefore, increase decision deferral. On the other hand, option refusal is an avoidance option that possibly closes the decision process, why it should be associated with a high NCC. In conclusion, NCC is additionally included as a personal variable within the correlational analyses to explore the association with the two different forms of decision avoidance.

**Further control variables.** A general derivation of previous studies is the assumption that difficult decisions increase deferral because they produce psychological conflicts, negative emotions, and uncertainty of preference [1,28,30,44,56]. Therefore, factors of the situation, as well as personal characteristics, should have a direct and indirect effect on decision avoidance, and the difficulty of a decision can be characterized as a mediator [1]. In this study, this mechanism is not the focus, and selection difficulty will not be systematically varied. Nevertheless, selection difficulty is an important factor of interest to influence decision avoidance and will, therefore, be included in the model as a predictor of decision deferral and option refusal. Selection difficulty will additionally be assessed as a dependent variable to evaluate the adequacy of this procedure.

The type of decision also has a significant impact on the handling of the situation [56]. Bigger and more important decisions in life are more complex and have greater consequences [13,36]. A correlative study showed that important decisions take more time than daily decisions [13]. Additionally, experimental study designs have shown decision importance to be a cue for deferral [57]. Therefore, a positive association between importance and decision deferral is expected from previous studies. However, this need not apply to option refusal as the inverse relationship can be assumed: If a decision is important, the costs of refusing options could be higher as in less important decisions. Analogous to selection difficulty, this factor will not be varied systematically but is included as a control variable in this study.

Furthermore, it is taken into account that option-sets with similarly attractive alternatives are more difficult [22,30] because decisions between similarly attractive alternatives produce psychological conflict. Thus, if there is no clear dominance of one option, decision avoidance increases [22,30]. It is expected that the similarity influences decision deferral as well as option refusal, but previous studies analyzed them separately. Similarity should increase decision deferral because more deliberation is needed to grasp subtle differences, but high similarity could increase option refusal as well if a decision-maker seeks for the one option that outshines others. Therefore, the similarity of alternatives will also be included as a control variable in this study. In sum, the following three variables will be captured as important factors and control variables, but not systematically varied: selection difficulty, the importance of a decision, and similarity of alternatives.

## Subjective decision-making quality: Post-decisional satisfaction

Classic decision-making research analyzes the act of choice under conditions of given options and full information. Thereby, the best choice can be determined. In realistic decision-making scenarios, however, there is no predefined best alternative, and the long-term consequences are unknown. Therefore, studies in naturalistic decision making are process-oriented without relying on normative choice models [58]. This study aimed to analyze how participants experience their avoidance behavior to capture functional and dysfunctional avoidance out of a subjective perspective instead of providing predefined best options. Thus, normative measurements to assess the quality of decision-making behavior (like the *multi-attribute utility theory* [59]) are not applicable in this context as the outputs are uncertain. Subjective measurements like the experienced utility [60] or emotions like regret [61] can be considered as alternative indicators for the quality of rational decision-making under uncertainty. Against this background, the *degree of satisfaction* with decision-making behavior is one suitable subjective measurement [62,63]. Inspired by the concept of choice-process satisfaction, it captures decision-making as well as decision avoidance [64]. The measurement provides a subjective evaluation, so that predefined probabilities and outcomes are not necessary.

Depending on personal and situational conditions, both strategies of decision deferral and option refusal could be either adaptive or dysfunctional avoidance: To defer a decision for further deliberation could be a reasonable step, e.g., to collect more information if there are enough recourses and less time pressure. On the other hand, extensive rumination, e.g., because of the fear of a wrong decision or negative consequences could be dysfunctional avoidance, if no information gain is expected. To refuse current options could be rational, if previous options are unattractive and one is not dependent on taking an option, e.g., staying at home and neither go to party A nor to party B, if both do not satisfy. However, refusing options can also be inappropriate or worsen the initial situation, e.g., if one refuses quite good job offers, despite one has financial problems. In this study, we assume that the decision-maker would be more satisfied with the choice of an avoidance option if he or she judges the

avoidance as rational rather than dysfunctional avoidance. Therefore, in this work, satisfaction with the decision-making behavior was used as a subjective indicator for the quality of decision-making behavior.

## The aim of the study and hypotheses

Overall, this study aimed 1) to investigate the impact of personal and situational factors that influence decision deferral and option refusal as two forms of decision avoidance and 2) to evaluate, the experienced satisfaction of either choosing, deferring, or refusing depending on manipulated context conditions. This study tests the hypotheses, if decision avoidance is increased under low time pressure, high lack of information, low attractiveness of the choice-set, and high personal indecisiveness. In there, it will be explored, if predictors differ between the two forms of decisional deferral and option refusal. Decision deferral should be increased by low time pressure, high lack of information, and high personal indecisiveness. Option refusal should especially be increased by the low attractiveness of the choice-set. Furthermore, the experienced satisfaction of either choosing, deferring, or refusing is explored depending on manipulated context conditions. The satisfaction serves as a subjective marker for the quality of the decision-making behavior to combine rational and emotional factors of decision avoidance in line with Anderson [1].

## Materials and methods

### Participants

Participants were recruited for an online study at Heidelberg University and other German universities by using students' e-mail distribution lists and forums as well as posters and flyers. They were informed about the aim and procedure of the study. Participants were included if they were students or doctoral candidates and completed the whole data set. Overall, n = 312 participants were analyzed within the study. Undergraduate psychology students of Heidelberg University received course credits for participation; other participants could win one of three 20€ Amazon vouchers. The study was carried out in accordance with the Declaration of Helsinki. All participants gave informed consent by beginning the online survey. The protocol was approved by the Ethics Committee of the Faculty of Behavioural and Cultural Studies at Heidelberg University.

### Study design

The experimental vignette study was conducted as a four-factorial mixed design. Three factors of the situation (*attractiveness of the choice-set*, *lack of information*, and *time pressure*) were manipulated two-staged in a within-subject design so that 2x2x2 = 8 different vignettes described one situation. Four different decision situations were varied in a between-subject design so that 4x8 = 32 vignettes were created. The importance of a decision, the similarity of the alternatives, and selection difficulty were expected to be influencing factors but were not systematically varied. Therefore, every participant received a subset of the vignettes that contained eight variations of one decision situation (e.g., all variations of the decision situation: seminar at university, see the first line in Table 1). This design enables a systematic analysis of main and interaction effects compared to a random selection of vignettes [65]. The behavior within the decision-making process was the primary dependent variable in this study. The satisfaction with the chosen decision-making behavior evaluated the quality. Implementation of the study was done within the software package 'SoSci Survey' [66].

**Table 1. Decision situations and the operationalization of variations across the three situational factors.**

| Decision situations | Basic description | Operationalizing of situational factors | | |
|---|---|---|---|---|
| | | Attractiveness | Lack of information | Time pressure |
| Seminar at university | You are compiling your study curriculum for the next semester, and you can choose between two seminars that are offered at the same time. | You can choose between a seminar with an **(un) motivated lecturer** and a seminar with an **(un) interesting topic**. | Examination modalities of both seminars are **unknown/(un)favorable**. | The enrolment period runs out **tomorrow/in a month**. |
| Plans for evening | You spend a weekend in your hometown, and you have two event invitations on Saturday evening. | You can choose between a **tedious/enjoyable family party** and a(n) **overcrowded/comfortable birthday party** of an old friend. | The exact course of both events is **unknown/(does not) please(s)** you. | The Saturday evening with both events takes place **tomorrow/in a month**. |
| Internship | You search an internship for the time between terms, and you have two different offerings. | You can choose between an internship with **(un) friendly colleagues** and **without/with a good payment** and an internship with **boring/varied activities** and **bad/good working hours**. | Your knowledge gain in both variants is **unknown/ low/high**. | The time between terms starts in a **week/in two months**. |
| Student apartment | You search a student apartment for the start of the semester, and you have two different offerings. | You can choose between an apartment that is **slightly over/exactly in your price category** and **located outside/near to the city center** and an apartment that is **slightly dilapidated/newly renovated** and **far from/near to the university**. | Neighbors of both apartments are **unknown/ pleasant/disturbing**. | The semester starts in a **week/in two months**. |

*Note*: The original material was provided in German. Vignettes were translated for publication. Each participant saw a subset of eight vignettes (2x2x2) within the same decision situation (one line within the table).

## Material

**Assessment of situational factors.** Decision situations were developed to represent common and meaningful situations in a student's life that could be described and systematically varied in vignette format. The following decision topics were chosen, partly inspired by other studies: Seminar at university [25,26], plans for the evening [22], student apartment [23,29,30], and internship. The vignette material is shown in Table 1.

Every situation is introduced with a short description (e.g., 'You are compiling your study curriculum for the next semester, and you can choose between two seminars that are offered at the same time.'). Subsequently, two alternatives are presented that vary between either two or four dimensions (e.g., lecturer and topic), depending on how big the decision is (four dimensions, big: student apartment and internship; two dimensions, small: seminar at university and plans for the evening).

To operationalize the factor attractiveness of the choice-set, the dimensions were accompanied by a description that reflected the attractiveness (e.g., motivated/unmotivated lecturer, interesting/uninteresting topic). This description was complemented by a further dimension (e.g., examination modalities) that was either unknown or presented with a valuation (favorable/unfavorable). This reflected the factor lack of information. Finally, time pressure was operationalized as high or low (e.g., the registration period is open until tomorrow/in a month). The time period was shorter in the smaller decision situations (tomorrow/in a month) than in the bigger decision situations (in a week/in two months). This structure was consistent across all vignettes so that only single words or clauses varied. The vignettes were evaluated in two steps to improve comprehensibility, realistic representation and intended manipulation (high/low time pressure, high/low lack of information, and high/low attractiveness of the choice-set).

**Assessment of personal factors.** Indecisiveness is deemed as the most important differential variable in decision deferral. The variable was operationalized as a decision-making style

(*decisional procrastination*) with the Melbourne Decision Making Questionnaire [48]. This scale has a good reliability (internal consistency: r = .81 [48]; retest-reliability after 1 month: r = .69 [67,68] and is recommended because of its validity [14]. Additionally, the *buck-passing style* (internal consistency: r = .87) was also included from the Melbourne Decision Making Questionnaire [48].

Further general decision-making styles out of the decision-making styles by Leykin and DeRubeis [6] were included to exploratory assess their association with decision avoidance behavior: confident (example item: 'I have faith in my decisions'; internal consistency: r = .92), rational ('My decision making requires careful thought'; internal consistency: r = .87), intuitive ('When I make decisions, I tend to rely on my intuition'; internal consistency: r = .68), and dependent decision-making syle ('I use the advice of other people in making my important decisions'; internal consistency: r = .87). Anxious decision-making was included as a psychopathological decision-making style ('I feel very anxious when I need to make a decision'; internal consistency: r = .90).

The original need for closure scale is 42 items long and consists of several subscales [47]. The German short scale of the NCC (Need for Cognitive Closure [69]) was used to minimize effort for participants. It consists of 16 items, had satisfying internal consistency, and has been validated in predicting the aversion of ambiguity in the Ellsberg Paradox [69].

## Procedure

The study procedure is visualized in Fig 1. After activating the link of the online survey, a welcome page asking for informed consent was shown. The topic of the study was introduced as *experiences and behavior in decision situations*, and the design of the vignette study was described. The study was divided into two parts: Part 1 contained the vignettes of decision situations and part 2 the questionnaires of personal factors. It was pointed out that one basic situation was shown with eight variants and that the changing parts were underlined. Participants were informed that they could either make a decision or decide to refuse or to defer choice, as they would do in reality.

With the "next"-button, part 1 started, and participants were randomly allocated to one of the four situations (without replacement). After reading one vignette, selection difficulty was rated by a one-item scale of 0–100. Afterward, the decision-maker could choose one of the two alternatives or choose an option of decision avoidance:

- I choose A. (decision A)

- I choose B. (decision B)

- I will decide at a later time. (*decision deferral*)

- I choose none of the alternatives. (*option refusal*)

The *degree of satisfaction* with the decision-making behavior was assessed after every decision with a one-item scale from 0–100: "I am satisfied with my behavior."

After completing the eight vignettes, the basic situation was shown again, and the importance of the decision and the similarity of the alternatives were rated by one-item scales of 0–100. To ensure external validity the following was captured: Participants were asked as to how realistically they experienced the decision situation and if they had their own experiences with a similar decision situation in the past (one-item scales of 0–100). Then in part 2, questionnaires and demographic data were assessed.

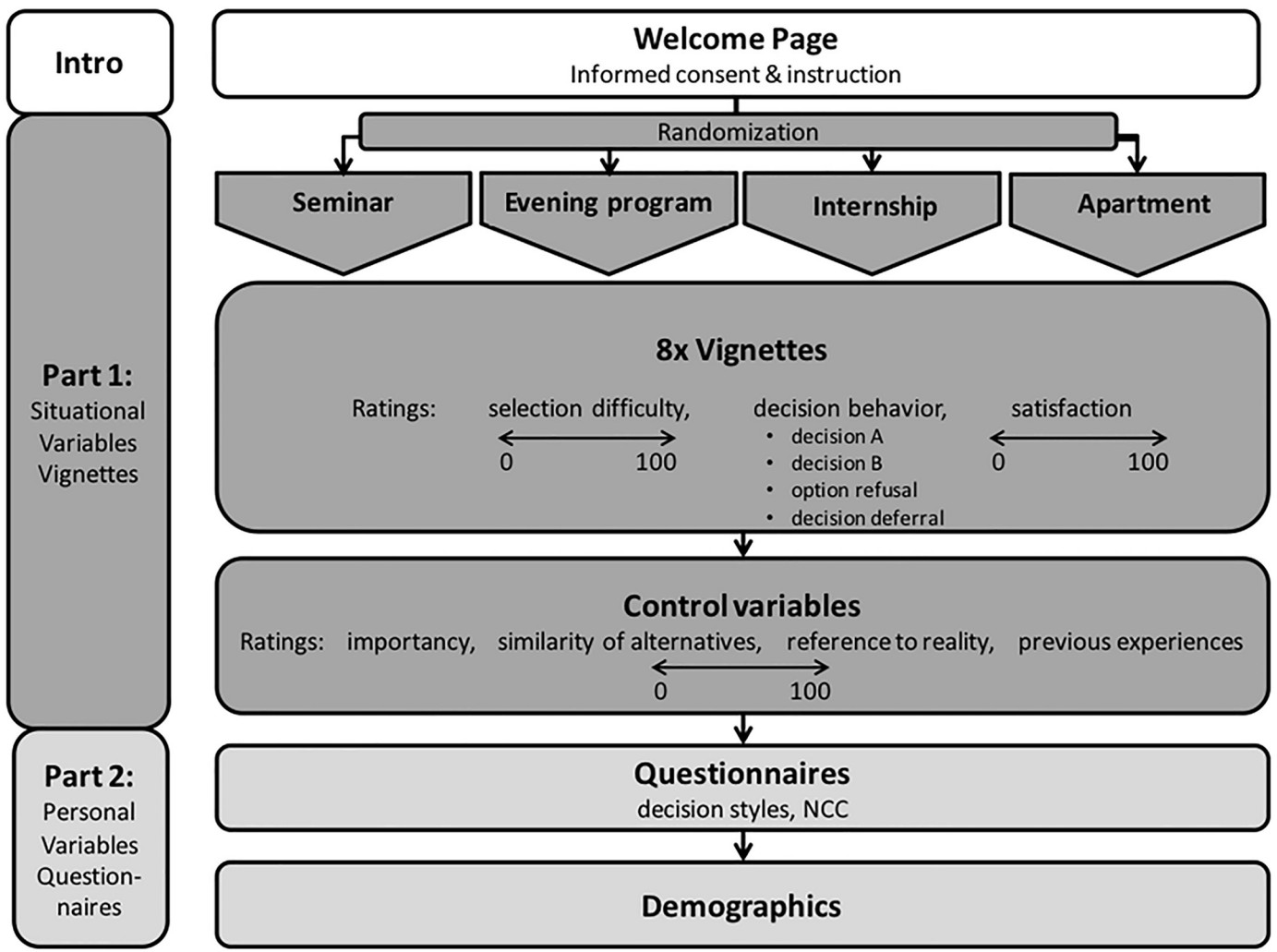

**Fig 1. Study procedure: Part 1 containing the eight vignettes, Part 2 the additional questionnaires and demographic data.**

### Statistical analyses

The main analyses were conducted with multi-level models to consider the hierarchic structure of the data due to repeated measurements. Every participant saw eight vignettes. Therefore, the situational factors in individual vignettes (level 1) were nested within general person-specific factors in questionnaire data (level 2). A random-intercept model was used that considers random intercepts across persons. The estimation of parameters was conducted with maximum-likelihood because the focus was on fixed effects instead of random variances [70]. The assumption of normal distribution was violated for some variables. This was considered negligible because of the big sample and the robustness of the model [71]. Multicollinearity was prevented by centering the predictors [72]. No imputation of missing values was necessary, thanks to the configuration of the survey (all questions had to be answered).

At first, a null model with a constant term and a random intercept was built to predict decision avoidance and the *Intraclass Correlation Coefficient* (ICC) was checked to determine the proportion of total variability in the outcome that is attributed to the person. ICC is used as a

measure of dependency of the data to check if a hierarchic method is appropriate. Afterward, the situational factors (time pressure, lack of information, and attractiveness of the choice-set), as well as the personal factor of indecisiveness and the control variables (selection difficulty, importance of a decision, and similarity of alternatives), were integrated. The coefficients within the multi-level models were interpreted analogous to linear or logistic regression analyses, respectively. Pseudo-$R^2$ was calculated to estimate the explained variance of decision avoidance by the model. The strengths of evidence are presented as odds ratios (OR) and evaluated based on Jeffreys' criteria [73]: OR = 0–1.6, barely worth mentioning; OR = 1.6–3.3, substantial; OR = 3.3–5.0, strong; OR = 5.0–6.6, very strong; OR>6.6, decisive. The effect of the decision situation was no primary focus in the study and therefore not integrated as a separate factor in the model. Nevertheless, descriptive results of the four decision situations with regard to variations in decision deferral and option refusal depending on the manipulated factors of the situation were provided. Additionally, correlation analyses were conducted to explore the association between personal factors and decision avoidance behavior. All statistical analyses were conducted using SPSS 21.0. To ensure data availability, the data set with all relevant variables in this study (S1 File), an overview of the variable labels, and the syntax of the multi-level models (S2 File) are provided in the supplement. For ethical reasons, sociodemographic factors are reported anonymous in a separated file, but this does not affect the reproducibility of the results.

## Results

### Sample characteristics

Of 782 participants that opened the online survey, 461 consented to participate and were randomized to one of the four subsets of vignettes. Of them, 316 completed the data sets. Four participants were excluded from analysis to ensure data quality. They indicated a low seriousness (< 50% on a scale 0–100) or the study was completed in such a short time (faster than five times the median of the processing time of all study participants) so that a detailed engagement with the material was questioned. Finally, N = 312 participants were included in the analysis; the study flow chart is shown in Fig 2. The participants rated the decision situations to be rather realistic (M = 70.13, SD = 22.28; scale 0–100). Some of the participants had already had their own experiences with a similar situation in the past (M = 45.82, SD = 30.88; scale 0–100). On average, decision situations were assessed as rather important (M = 70.60, SD = 22.76; scale 0–100) and moderately difficult (M = 42.00, SD = 21.18; scale 0–100).

Overall, 62.2% of the study participants were female, and the mean age was 23.8 (SD = 3.50, range 18–40). They were mainly students (93.6%), 62.8% were undergraduate students, 27.6% were psychology students or doctoral candidates in psychology.

### Decision avoidance behavior

The chosen decision-making behavior within the eight vignettes was distributed as follows: On average, six of eight decisions were made (M = 5.50, SD = 1.64), one decision was deferred (M = 1.03, SD = 1.36), and one decision was refused (M = 1.47, SD = 1.37). Half of the participants deferred no decision, 18.9% one, 13.8% two, and 9.6% three vignettes. Four to seven vignettes were deferred in fewer than 5% of cases, and no participant decided to defer all decisions. One third (34.6%) of the participants decided to refuse no decision, 16.3% refused one vignette, 28.8% two, 7.7% three, and 12.2% four vignettes. One person refused five vignettes (0.3%), and no person refused six or more decisions. Overall, 16.7% of the participants never used an option of decision avoidance, 17.9% only used the option to defer a decision at least

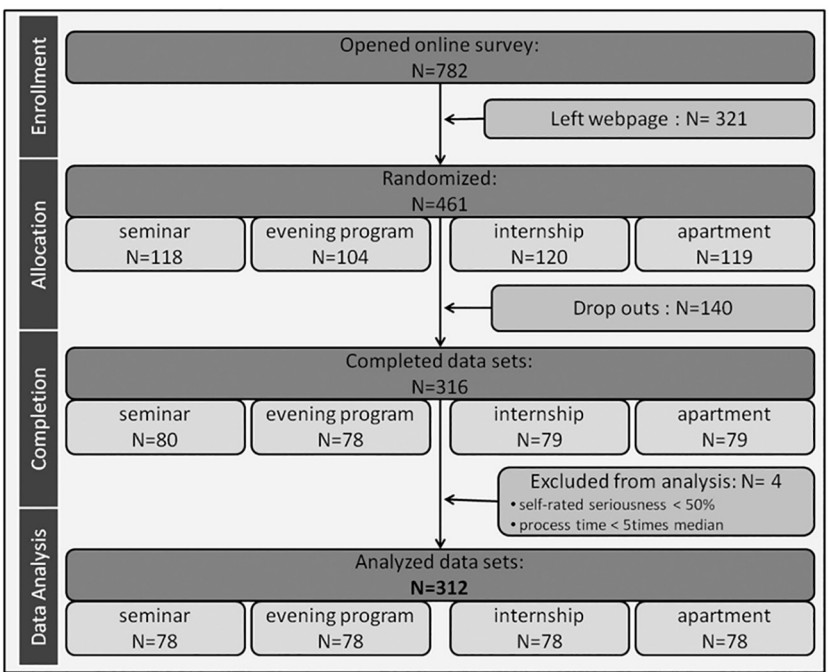

**Fig 2. Study flow chart: Reduction process of N = 782 participants that opened the online survey to N = 312 that completed the study and were analyzed finally (with N = 78 for each vignette).**

once, 34.9% only used option refusal at least once, and 30.4% used both decision deferral and option refusal depending on the vignette.

Four different decisional situations adapted to a student's everyday life had been studied: seminar at university, plans for the evening, internship, and student apartment. Decision deferral was most frequent when deciding on plans for the evening (M = 1.53, SD = 1.59) and least frequent when deciding on an apartment (M = 0.53, SD = 0.77). Option refusal was most frequent when deciding between two internships (M = 2.49, SD = 1.49) and least frequent when deciding on plans for evening (M = 0.62, SD = 0.91). To evaluate the effect of the within-subject manipulated factors, the four situations were aggregated to increase generalizability.

Below, the personal and situational factors will be investigated regarding their influence on decision deferral and option refusal as different forms of decision avoidance.

### Situational factors

Regarding the situational factors, the dimensions (low/high) of manipulated time pressure, lack of information, and attractiveness of the choice-set (x-axis) were regarded as dependent on their probability of decision deferral and option refusal (y-axis: 0–100%), as shown in Fig 3. In there, results were aggregated for the four different decision situations. To get an overview of the four different decision situations and the interactions of the manipulated factors, see Table 2. In the following, results were presented according to hypotheses, but the final evaluation will be given according to multi-level models:

Time pressure: According to H1, it was assumed that low time pressure is associated with more decision avoidance. Decision deferral was more likely in the condition of low time pressure (24%) than in the condition of high time pressure (2%), while option refusal was less influenced by time pressure (Fig 3). Therefore, the expected effect was found in both forms of

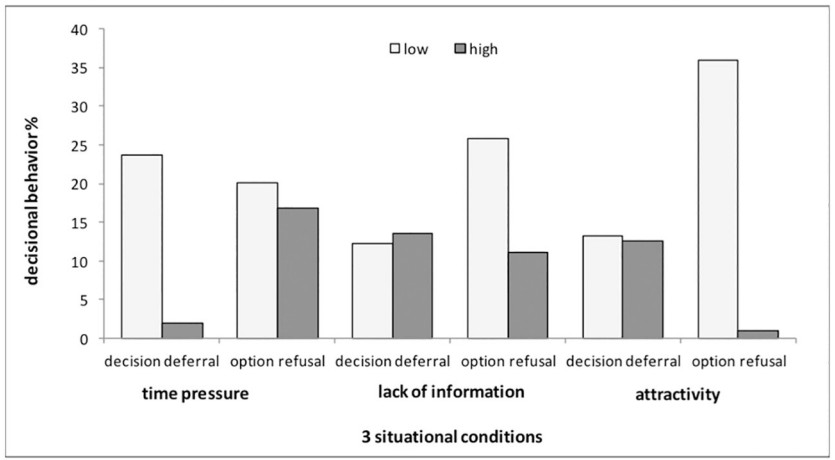

**Fig 3. The three situational factors "time pressure", "lack of information," and "attractivity" of the choice-set (white: low; dark: high) and the respective probabilities of decision deferral and option refusal (in percent).**

decision avoidance, but especially for decision deferral. By descriptively evaluating the four decision situations separately (Table 2), this effect appears to be consistent across the four situations.

Lack of information: H2 expected a high lack of information to be associated with more decision avoidance. However, overall, high or low lack of information had hardly any effect on decision deferral, while option refusal was higher with a low lack of information (26%) than with a high lack of information (11%) (Fig 3). The results for option refusal were not expected. By looking at the different combinations (Table 2), only under the condition of an attractive choice-set and low time pressure, a high lack of information (code 221) implied more decision deferral than a low lack of information (code 211). This effect seems to be consistent across the four situations (Table 2).

Attractiveness of the choice-set: According to H3, decisions with an unattractive choice-set were expected to result in more decision avoidance. In our study, the attractiveness of

**Table 2. Variations in decision deferral and option refusal depending on the manipulated factors of the situation.**

| Coding (AIT) | Overall (N = 312) | | Situation 1 seminar (n = 78) | | Situation 2 plans for evening (n = 78) | | Situation 3 internship (n = 78) | | Situation 4 apartment (n = 78) | |
|---|---|---|---|---|---|---|---|---|---|---|
| | Decision deferral | Option refusal | Decision deferral | Option refusal | Decision deferral | Option refusal | Decision deferral | Option refusal | Decision deferral | Option refusal |
| 111 | 25.32 | 51.60 | 37.18 | 37.18 | 35.90 | 21.79 | 17.95 | 70.51 | 10.26 | 76.92 |
| 112 | 3.21 | 49.36 | 0.00 | 51.28 | 6.41 | 29.49 | 5.13 | 74.36 | 1.28 | 42.31 |
| 121 | 22.12 | 26.28 | 25.64 | 8.97 | 30.77 | 1.28 | 8.97 | 57.69 | 23.08 | 37.18 |
| 122 | 2.24 | 16.35 | 2.56 | 12.82 | 2.56 | 3.85 | 3.85 | 42.31 | 0.00 | 6.41 |
| 211 | 19.23 | 1.28 | 25.64 | 0.00 | 35.90 | 1.28 | 10.26 | 1.28 | 5.13 | 2.56 |
| 212 | 1.28 | 0.96 | 2.56 | 2.56 | 1.28 | 1.28 | 1.28 | 0.00 | 0.00 | 0.00 |
| 221 | 28.21 | 0.96 | 32.05 | 0.00 | 39.74 | 1.28 | 28.21 | 1.28 | 12.82 | 1.28 |
| 222 | 1.28 | 0.64 | 2.56 | 0.00 | 0.00 | 1.28 | 2.56 | 1.28 | 0.00 | 0.00 |

*Note*. The coding concerns the influencing factors of attractiveness (A), lack of information (I) and time pressure (T), whereby 1 represents a low level and 2 a high level; values are presented in %.

alternatives had hardly any effect on decision deferral, either, but showed the biggest difference regarding option refusal. Decision situations with two attractive options were only refused in 1% of cases, while 36% refused options when both were unattractive (Fig 3). This effect seems to be approximately consistent across the four situations, but even more salient in the bigger decision situation 3 (internship) and 4 (apartment) than in decision situation 1 (seminar) and 2 (plans for evening) (Table 2).

By regarding option refusal within the single vignettes, we recognized an interaction effect, so that a low lack of information implied higher option refusal, only combined with low attractiveness. Therefore, the interaction was integrated into the multi-level model to evaluate hypotheses finally.

## Personal factors

Regarding the personal factors, indecisiveness, further individual decision-making styles, and the NCC were correlated with the frequency of decision deferral and option refusal (variable coded as 0–8), as shown in Table 3.

Indecisiveness: According to H4, it was assumed that the higher the indecisiveness of a person, the more decision avoidance is shown. Indeed, the correlation analyses showed that higher indecisiveness was associated with more decision deferral, but with less option refusal. Therefore, from these results, this hypothesis can only be accepted for decision deferral. Additionally, indecisiveness was correlated with selection difficulty ($r = .24$, $p < .05$). A final assessment follows below because indecisiveness as the most important factor was analyzed within the multi-level models.

Decision-making styles: According to the correlation analyses, only lower confidence was associated with decision deferral, and a higher intuitive style was associated with option refusal, conform to the expectations. Nevertheless, associations were small, and most of the decision-making styles showed no association with decision avoidance behavior. Instead, confident ($r = -.26$, $p < .05$), buck-passing ($r = .20$, $p < .05$), dependent ($r = .19$, $p < .05$), and anxious decision-making style ($r = .26$, $p < .05$) were associated with selection difficulty.

NCC: In our analysis, the NCC score showed neither an association with decision deferral nor with option refusal. Instead, a high NCC was associated with the experience of higher selection difficulty ($r = .17$, $p < .05$).

**Table 3. Correlation analyses about the associations between personal factors, selection difficulty, and the two forms of decision avoidance.**

| Personal Factors | Decision avoidance | |
|---|---|---|
| | Decision deferral | Option refusal |
| Indecisiveness | .21* | -.12* |
| Buck-passing style | .10 | -.09 |
| Confident style | -.17* | .07 |
| Rational style | -.04 | .04 |
| Intuitive style | -.01 | .11* |
| Dependent style | .06 | -.02 |
| Anxious style | .10 | -.06 |
| NCC | .06 | -.02 |

*Note*: NCC = need for cognitive closure; Correlation between the personal characteristics and frequency of decision avoidance, level of significance, $p < .05$ *, two-sided.

## Multi-level-models

For final evaluation, multi-level-models were used to consider the hierarchic structure of the data (situational factors and personal factors). The models were built separately for decision deferral and option refusal. According to the random-intercept model, a constant term was used and integrated as fixed and random effect.

The first model aimed to predict decision deferral. At first, the null model with a constant term and variable intercept were built. ICC showed that 92.2 percent of variance could be attributed to factors within a person, and 7.8 percent of variance was due to stable interindividual differences. This justified the choice of a random-intercept model. Afterward, predictors were integrated. Therefore, the manipulated situational factors (time pressure, lack of information, and attractiveness of the choice-set), the personal factor of indecisiveness, and the control variables (selection difficulty, the importance of a decision, and similarity of alternatives) were integrated. Additionally, a second model was built that integrated the interaction lack of information*attractiveness. As model fit increased by adding the interaction to the previous model $\chi^2(1) = 1056.09–1024.50 = 31.59$ ($> \chi^2$crit $= 3.84$), the interaction was kept; results are shown in Table 4.

The model fit of the complete model compared to the model with the constant term was $\chi^2(8) = 1555.54–1024.50\ 645 = 531.04$ ($> \chi^2$crit $= 15.51$); the baseline-error variance of the model with the constant term was $\sigma^2 = 0.11$, after adding the influencing factors it was $\sigma^2 = 0.08$ (Pseudo-$R^2 = 0.23$). Therefore, all predictors together explained 23.4% of the variance of the deferral behavior.

For analyzing option refusal, a second model was built on the same basis. Model fit increased by adding the interaction to the previous model, $\chi^2(1) = 1316.56–1141.30 = 175.26$ ($> \chi^2$ crit $= 3.84$) as well. The model fit of the complete model compared to the model with the constant term was $\chi^2(8) = 2244.26–1141.30 = 1102.96$ ($> \chi^2$crit $= 15.51$); the baseline-error variance of the model with the constant term was $\sigma^2 = 0.15$, after adding the influencing factors it was $\sigma^2 = 0.09$ (Pseudo-$R^2 = 0.43$). Therefore, all predictors together explained nearly half of the variance (42.9%) of the refusal behavior.

Now the hypotheses of situational predictors and individual predictor will be regarded:

**Table 4. Results of the multilevel models of decision deferral and option refusal.**

| | Dependent factors of decision avoidance | | | | | | | | | |
| --- | --- | --- | --- | --- | --- | --- | --- | --- | --- | --- |
| | Decision deferral | | | | | Option refusal | | | | |
| Parameters | Coef. | F value | P value | OR | 95% CI (OR) | Coef. | F value | P value | OR | 95% CI (OR) |
| Constant Term | -5.24 | 35.032 | p<.001 | | | -6.02 | 47.869 | p<.001 | | |
| Time pressure (= 0) | 3.87 | 220.44 | p<.001 | 48.09 | 28.82, 80.17 | 1.01 | 39.840 | p<.001 | 2.73 | 2.00, 3.74 |
| Lack of information (= 0) | -0.54 | 5.671 | p = .015 | 0.59 | 0.38, 0.90 | 0.22 | 13.515 | p = .712 | 1.25 | 0.39, 4.02 |
| S Attractiveness (= 0) | 0.14 | 33.058 | p<.001 | 1.15 | 0.73, 1.81 | 4.13 | 227.432 | p<.001 | 61.93 | 23.95, 160.25 |
| Attractiveness (= 0) * lack of information (= 0) | 1.86 | 31.125 | p<.001 | 6.43 | 3.34, 12.37 | 1.84 | 8.787 | p = .003 | 6.29 | 1.86, 21.22 |
| P Indecisiveness | 0.28 | 5.332 | p = .021 | 1.32 | 1.04, 1.67 | -0.14 | 1.431 | p = .232 | 0.87 | 0.69, 1.09 |
| Selection difficulty | 1.02 | 94.988 | p<.001 | 2.78 | 2.26, 3.41 | -0.07 | 0.534 | p = .465 | 0.94 | 0.78,1.12 |
| C Importance | 0.08 | 0.570 | p = .476 | 1.09 | 0.86, 1.37 | -0.29 | 7.234 | p = .007 | 0.75 | 0.61,0.93 |
| Similarity | 0.13 | 1.292 | p = .256 | 1.14 | 0.91, 1.42 | 0.69 | 36.920 | p<.001 | 2.00 | 1.60, 2.50 |

*Note*: The table shows the model parameters (S = situational factors, P = personal factor, C = control variables) with coefficients (Coef.), F-values, p values, and Odds Ratios (OR), separated for decision deferral and option refusal. OR>6.6, according to Jeffreys (73) criteria, were shown in bold. All denominator degrees of freedom (df) in the model of decision deferral and option refusal were 2.027 and 2.166, respectively.

Time pressure: The odds ratio of decision deferral was 48-times higher with low time pressure than with high time pressure, $F_{(2.027)} = 220.447$, $p < .001$ (OR = 48.09). Option refusal was also more likely with low time pressure, but the effect was clearly smaller, $F_{(2.166)} = 39.840$, $p < .001$ (OR = 2.73). Therefore, the hypothesis can be accepted for both forms of decision avoidance but seems to be especially relevant for decision deferral.

Lack of information: The probability of decision deferral was higher with a high lack of information than with a low lack of information, $F_{(2.027)} = 5.671$, $p = .015$ (OR = 0.59). Lack of information had no effect on option refusal, $F_{(2.166)} = 13.515$, n.s. Therefore, the hypothesis can only be accepted for decision deferral.

Attractiveness of alternatives: Decisions between unattractive alternatives were more likely to be deferred than decisions between attractive alternatives, $F_{(2.027)} = 33.058$, $p < .001$ (OR = 1.15). This effect was clearer with low lack of information, $F_{(2.166)} = 31.125$, $p < .001$ (OR = 6.43). The odds ratio of option refusal was 62-times higher within decisions between unattractive options-sets than between attractive option-sets, $F_{(2.027)} = 227.432$, $p < .001$ (OR = 61.93). This effect was smaller, but also relevant in combination with low lack of information, $F_{(2.166)} = 8.787$, $p = .003$ (OR = 6.29).

Indecisiveness: Within the multi-level models, indecisiveness was a predictor for decision deferral, so that higher indecisiveness implied a higher probability to defer choice, $F_{(2.027)} = 5.332$, $p = .021$ (OR = 1.32), but not for option refusal, $F_{(2.166)} = 1.431$, n.s. Therefore, the hypothesis can only be accepted for decision deferral.

## Subjective decision-making quality: Post-decisional satisfaction

The satisfaction with the individual decision-making behavior was assessed to capture a subjective measure for the quality of decision-making behavior. Results are presented overall conditions and separated for the different conditions (see Table 5).

It should be noted that participants were most discontent if they deferred the decision, while the satisfaction with a taken decision was comparable to a refused decision. An exception was the condition of unattractive options, where satisfaction was best if both options were refused.

## Difficulty analysis

Selection difficulty was supposed to be a predictor of decision avoidance behavior and therefore integrated as a control variable within the multi-level-models. According to the conflict theory [44], difficult decisions increase deferral. Furthermore, several personal and situational factors are supposed to increase selection difficulty so that it should be a mediator [1]. Therefore, we did an additional analysis, where we considered selection difficulty as a dependent

**Table 5. Satisfaction with the decision-making behavior overall and separately for the manipulated situational conditions.**

| Behavior | Overall | | Time pressure = 0 | = 1 | Lack of information = 1 | Conditions | | |
|---|---|---|---|---|---|---|---|---|
| | | | | | | = 0 | Attractive-ness = 0 | = 1 |
| Decision taken | 70.45 (26.97) | | 73.14 (24.43) | 70.51 (27.33) | 70.40 (24.81) | 77.28 (22.59) | 58.84 (27.20) | 68.58 (26.84) |
| Option refusal | 73.33 (27.91) | | 76.06 (26.49) | 70.09 (29.25) | 68.26 (27.31) | 75.51 (27.93) | 73.56 (27.94) | 64.75 (26.46) |
| Decision deferral | 54.75 (29.56) | | 55.18 (28.83) | 49.68 (37.48) | 54.90 (30.53) | 54.58 (28.56) | 54.29 (30.56) | 55.23 (28.56) |

*Note*: The table shows the satisfaction with the decision-making behavior with means and standard deviations; overall and separately for the manipulated conditions (0 = low, 1 = high). In the following conditions, decision avoidance was supposed to be increased: Low time pressure, high lack of information, low attractiveness.

**Table 6. Results of the multilevel models of selection difficulty.**

|   | Parameter | Coefficient | *F*-value | *df* | p value |
|---|---|---|---|---|---|
|   | Constant Term | 40.39 | 724.734 | 872.503 | p < .001 |
| S | Time pressure | 3.67 | 10.969 | 2183.622 | p = .001 |
|   | Lack of information | 4.78 | 18.671 | 2183.622 | p < .001 |
|   | Attractiveness | -5.23 | 18.650 | 2416.181 | p < .001 |
| P | Indecisiveness | 4.81 | 18.142 | 312.432 | p < .001 |
| C | Similarity | 5.12 | 39.183 | 1763.134 | p < .001 |
|   | Importance | 4.31 | 25.073 | 1383.648 | p < .001 |

*Note*: The table shows the model parameters (S = situational factors, P = personal factor, C = control variables) with coefficients, F-values, denominator degrees of freedom (df), p values; Model fit increased by adding the interaction to the previous model $\chi^2(6) = 24209.68–24099.20 = 110.48$ ($>\chi^2$krit = 12.59). The baseline-error variance of the model with the constant term was $\sigma^2 = 790.57$, after adding the influencing factors it was $\sigma^2 = 764.51$ (Pseudo-$R^2 = 0.03$).

variable (Table 6). As Pseudo-$R^2$ only explained 3% of the variance, this direction was negligible in our study.

## Discussion

This study aimed 1) to investigate the impact of personal and situational factors that influence decision deferral and option refusal as two forms of decision avoidance and 2) to evaluate the experienced satisfaction of either choosing, deferring, or refusing depending on manipulated context conditions. In summary, decision deferral was increased by situational factors (low time pressure, high lack of information, and low attractiveness of the choice-set), higher selection difficulty, and higher indecisiveness of the decision-maker. The refusal of current options was increased by situational factors (low attractiveness of the choice-set, lower importance, and higher similarity of alternatives). Post-decisional satisfaction was higher after option refusal than after decision deferral. In the following, results are discussed according to hypotheses:

### Situational factors

At first, situational predictors were considered. It was assumed that low time pressure increases decision avoidance. Indeed, this was confirmed for decision deferral as well as for option refusal. Thereby, the effect of time pressure on decision deferral was considerably greater and the strongest predictor in this study (OR = 48). The results emphasize the fact that people adapt their decision-making behavior to the time frame [74]. Furthermore, the direction is in line with previous studies, in which higher time pressure decreased the probability of avoidance behavior [23,30,36]. To defer choice under conditions of low time pressure could be an adaptive strategy to improve the understanding of the situation [13]. Independently of the expected gain of information, deferral has its added value by increasing certainty with decision and acceptance of unpleasant decisions [35,36]. Furthermore, option refusal in the condition of low time pressure can be useful to search for further options.

According to the second hypothesis, it was assumed that a lack of information increases decision avoidance. This could be confirmed for decision deferral, but not for option refusal. It was assumed that a lack of information increases uncertainty, and therefore increases deferral because uncertainty could be reduced by collecting information [20,38,39]. Nevertheless, the effect in this study was less decisive than other situational factors (OR = 0.59). This could be due to the manipulation because only a single piece of information was missing in the 'high

lack of information' condition. In further studies, it would be interesting to explore whether a more significant variation of uncertainty strengthens this effect.

The third hypothesis implied that decisions between unattractive alternatives were more often avoided than decisions between attractive alternatives. This was confirmed for decision deferral as well as for option refusal. Here, the effect on decision deferral was rather small (OR = 1.15), but the effect on option refusal was very decisive (OR = 62). This result is in line with a previous study, in which no alternative was chosen in the condition of an unattractive choice-set [27]. In this condition, it seems reasonable to actively search for new alternatives [20,75] or to passively wait for improved or new options to emerge [26,39]. It is possible that the effect on decision deferral would have been larger if option refusal had not been possible, as it is sometimes the case in reality.

Independently from hypotheses, the condition of low lack of information and low attractiveness of the choice-set (interaction effect) increased decision deferral and option refusal. This was due to the manipulation, because the additional information was presented with a valuation that simultaneously strengthened the condition of attractiveness. Thus, this result is expected to be due to low attractiveness and not due to a low lack of information. In further studies, it is necessary to separate the two conditions and insert a neutral description of the lack of information condition.

## Personal factors

According to personal factors, a higher indecisiveness of a person was expected to be associated with more decision avoidance. This was only confirmed for decision deferral, but not for option refusal. The effect on decision deferral was consistent with previous studies [52,54,76]. Nevertheless, the effect was rather small, and context conditions were more decisive than personal factors. Further studies should explore if decision situations of higher complexity and risk manipulation increase the effect. In this regard, a behavioral process study of Ferrari and Dovidio [25] showed that indecisive persons indeed only show increased decision deferral under the condition of high risk.

According to several decision-making styles, analyses were exploratory, but an association with decision avoidance was assumed. Indeed, only lower confidence was associated with decision deferral, and a higher intuitive style was associated with option refusal. The association between lower confidence and decision deferral was in line with the conflict theory of decision-making [44]. This is suggested to be a somewhat maladaptive form of decision-making. Nevertheless, deferral can increase certainty and thus increase the likelihood of implementing a decision into action, whereby deferral can be an adaptive behavior. The association between intuitive decision-making and option refusal suggests option refusal to be a fast and emotion-based behavior, which has to be clarified in further studies. It is, however, an interesting result that neither decision deferral nor option refusal was associated with anxious decision-making as could have been expected. Furthermore, option refusal was not associated with buck-passing and even negatively correlated with deferral style. This suggests that option refusal is an adaptive behavior and is not associated with avoiding responsibility for a decision or chronic indecisiveness. Overall, decision-making styles showed a higher association with selection difficulty than with decision-making behavior. Therefore, it is necessary to make this distinction in future studies.

A high NCC was expected to reduce decision avoidance, but the NCC showed neither an association with decision deferral nor with option refusal. According to the definition of the construct, an association was expected because a high NCC describes the tendency to accelerate the decision-making process by searching for less information or contradictions [47]. A

possible explanation for the lacking connection could be the heterogeneity of the subscales. Neuberg, Judice, and West [77] showed that the subscales preference for predictability, preference for order, and discomfort with ambiguity are highly correlated, whereas the subscale closed-mindedness is not correlated with the other subscales and the subscale decisiveness is negatively correlated. In this study, a factor analysis also showed a heterogeneous structure (data not reported here), so that opposite effects could have affected avoidance behavior that further studies have to specify.

## Control variables

Besides the hypotheses, selection difficulty, the importance of a decision, and similarity of alternatives were integrated as control variables. A general derivation of previous studies was the assumption that difficult decisions increase deferral [1,28,30,44,56]. In this study, it was confirmed that decisions that were experienced as more difficult, were deferred more often (OR = 2.78), while no connection with option refusal was found. Furthermore, the importance of a decision was expected to increase deferral, because previous studies have shown that more important decisions take more time to deliberate [13] and imply increased deferral [57]. This was not confirmed because no association was found between importance and deferral. Instead, the opposite direction was found for option refusal (OR = 0.75). Additionally, it was expected that choice-sets with similarly attractive alternatives are more difficult [22,30] and, therefore, more often deferred. This effect was not confirmed for deferral, but for option refusal in this study. Overall, it has to be mentioned that these control variables were not systematically varied, wherefore the results should be interpreted with caution.

## Decisional satisfaction

In addition to the factors that influence decision avoidance, it was the second aim of this study to investigate the adaptivity of the different forms of decision avoidance. Therefore, decisional satisfaction was captured as a subjective indicator of the quality of decisional behavior as different authors have suggested [62–64]. Interestingly, participants were similarly satisfied with a decision that had been made and a decision that had been refused, whereas satisfaction was lowest after a deferred decision. The higher satisfaction after option refusal, in contrast to decision deferral, could be explained by the achieved clarification and reduction of uncertainty that is maintained during the deferral. Furthermore, the frequency of deferred decisions was correlated with lower individual confidence in the study. So it could be assumed that participants with lower confidence are also less satisfied with their decisional behavior. A special case was the condition of low attractiveness, in which satisfaction was highest after option refusal and satisfaction after a taken decision was as low as after deferral. The focus after a refused decision during an avoidance-avoidance conflict is a positive one, whereas it is negative when a decision has been made, which is a possible explanation for the differences in satisfaction [78]. Overall, the results provide a first hint that option refusal provides more satisfaction than decision deferral and could, therefore, be a more adaptive behavior, especially in the case of an unattractive option set. Nevertheless, long term consequences and satisfaction have yet to be captured in future studies to assess the subjective quality of a decision finally.

## Limitations and strengths

This study takes up some concerns that naturalistic decision-making research has about classic decision-making research (e.g., [58]): The vignettes reflect familiar decisional situations that are formulated in a realistic way and fit our sample of student participants. Additionally, the study focuses on process-orientation in contrast to input-output-orientation by analyzing

when people chose an avoidance option and how satisfied they are with it instead of providing predefined best choices. Nevertheless, this study evaluated hypothetical decision situations and no real-life behavior to systematically vary different dimensions to evaluate the importance of the prediction of an avoidance option. With this approach, we are in line with Todd and Gigerenzer [32] that argue no general contradiction between formal modeling and descriptively valid and task-specific real-world decisions. Therefore, we consider important real-life context conditions like time pressure and incompleteness of the information but formalize them in standardized vignettes to combine different advantages.

Several limitations have to be considered. The present study was conducted with self-developed vignettes so that results could deviate from real decision-making behavior [79]. It can be assumed that decision avoidance is of greater importance in real-life decision situations and the possibility of socially desirable responses has to be considered because decision deferral is seen a weakness. Nevertheless, vignette studies provide high internal validity and allow to hierarchically assess personal and situational factors [65]. A further limitation is that a manipulation check of the situational variables was evaluated before the start of the study but not integrated during the study so that it is not clear if e.g., one month is equally considered as low time pressure by every participant. Additionally, there may be differences in the salience of vignette manipulation. Therefore, the effects of the manipulated factors cannot be compared so easily. Besides, personal factors were largely examined in a correlative manner, and only indecisiveness was integrated into the final model. It could be assumed that personal factors gain importance if more extreme samples like clinical ones were considered. Finally, decision deferral and option refusal are different forms of decision avoidance but not necessarily have to be distinct. However, our results showed that participants seem to experience the two forms as qualitatively different because there were clear intraindividual variations in the use of the form of avoidance option depending on situational variations in the vignettes.

The study has several strengths: Decision avoidance is an important topic that has been studied from different, but unlinked psychological disciplines. In contrast to other studies, personal as well as situational factors were considered, so that different perspectives and psychological disciplines were integrated. Additionally, two different avoidance options were considered, which is a great advantage over previous studies [1]. This study is in line with current developments in decision-making research and assessed the satisfaction with decision avoidance to capture subjective functionality instead of predefined best alternatives like in classic decision-making research. Furthermore, four different decision situations with two smaller and two bigger decisions raise the possibility of generalization. Finally, decision situations were described in a realistic way, and stimulus material and student sample constituted a good fit. This was also supported by our participants that assessed the decision situations to be rather realistic. Overall, this strengthens external validity.

There are several recommendations for future studies. Besides personal and situational factors, the interaction of person and situation or different situational factors could be integrated by including mediators or moderators. For example, time pressure is assumed to be a moderator of other factors, so that some situational factors only influence decision avoidance in the condition of low time pressure but not in the condition of high time pressure. Furthermore, it would be interesting to integrate further manipulation steps to evaluate when a factor becomes significant for decision avoidance. Finally, a more extensive selection of decision situations and a third avoidance option of buck-passing could be implemented. To evaluate the specific effects of the decision situations further studies could integrate them as a separate factor within the multilevel models.

## Conclusion

To sum up, decision deferral was affected by situational factors, selection difficulty, and indecisiveness of the decision-maker. In the manipulation of this study, time pressure appears to be the strongest predictor (OR = 48). The refusal of current alternatives was exclusively influenced by situational factors, especially the manipulated attractiveness of the choice-set (OR = 62). Therefore, option refusal seems to be an adaptive strategy that is adjusted to situational conditions. On the other hand, decision deferral partly depends on maladaptive personality factors like indecisiveness and confidence. Nevertheless, time pressure had by far the largest effect on deferral, which is considered adaptive as well. Post-decisional satisfaction was higher after option refusal than after decision deferral, which is supposed to be an indicator of quality. Overall, it is important to broaden the view on decision avoidance and distinguish between adaptive and maladaptive forms. To implement this, future studies on decision-making have to include different options of decision avoidance and rethink their measures for the quality of decision-making.

## Supporting information

**S1 Dataset. Predictors of decision avoidance.**
(XLSX)

**S1 File. Overview of variables in the dataset.**
(DOCX)

**S2 File. Syntax multi-level mode.**
(TXT)

## Author Contributions

**Conceptualization:** Sabrina Berens.

**Data curation:** Sabrina Berens.

**Investigation:** Sabrina Berens.

**Supervision:** Joachim Funke.

**Writing – original draft:** Sabrina Berens.

**Writing – review & editing:** Joachim Funke.

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
