## [Decision Letter · Decision Letter 0]

29 Aug 2019

PONE-D-19-17779

A Vignette Study of Option Refusal and Decision Deferral as two Forms of Decision Avoidance: Situational and Personal Predictors

PLOS ONE

Dear Dr. Funke,

Thank you for submitting your manuscript to PLOS ONE. After careful consideration, we feel that it has merit but does not fully meet PLOS ONE’s publication criteria as it currently stands. Therefore, we invite you to submit a revised version of the manuscript that addresses the points raised during the review process.

Please find below the reviewers' comments.

We would appreciate receiving your revised manuscript by Oct 13 2019 11:59PM. To enhance the reproducibility of your results, we recommend that if applicable you deposit your laboratory protocols in protocols.io, where a protocol can be assigned its own identifier (DOI) such that it can be cited independently in the future. For instructions see: http://journals.plos.org/plosone/s/submission-guidelines#loc-laboratory-protocols

We look forward to receiving your revised manuscript.

Kind regards,

Valerio Capraro

Academic Editor

PLOS ONE

Journal Requirements:

Additional Editor Comments (if provided):

I have now collected three reviews from three experts in the field. The reviewers are somehow split, with one recommending rejection and two recommending major revisions. After reading the reviews and the manuscript, I feel that a revision of this manuscript could be publishable. Therefore, I would like to invite you to revise your paper according to the reviewers' comments. Needless to say that all comments must be addressed, especially those from the negative reviewer.

Reviewers' comments:

Reviewer's Responses to Questions

**Comments to the Author**

1. Is the manuscript technically sound, and do the data support the conclusions?

Reviewer #1: Partly

Reviewer #2: No

Reviewer #3: Partly

2. Has the statistical analysis been performed appropriately and rigorously? 

Reviewer #1: Yes

Reviewer #2: No

Reviewer #3: I Don't Know

3. Have the authors made all data underlying the findings in their manuscript fully available?

Reviewer #1: Yes

Reviewer #2: Yes

Reviewer #3: No

4. Is the manuscript presented in an intelligible fashion and written in standard English?

Reviewer #1: Yes

Reviewer #2: No

Reviewer #3: No

5. Review Comments to the Author

Reviewer #1: Thank you for providing me with the opportunity to review your manuscript. I found it an interesting read and enjoyed reviewing it. I do however have some significant reservations with regards to theory and design, which means I am recommending major revisions. I have two main points. First, I found the theoretical background to the paper was under-developed and confused in places. I wasn’t entirely sure how your paper was unique or what new questions it was addressed. This is not because I believe it lacks these points, I just feel it wasn’t conveyed clearly in the writing. Second, I was unsure of the qualitative difference between choice deferral and option refusal. Did these options actually result in different consequences for participants, or were they both effectively the same in terms of consequences for the participant (i.e., they didn’t make a decision)? I found it difficult to evaluate your findings/discussion as I wasn’t convinced by these options being distinct enough, especially in terms of considering whether participants perceived these options as different or the same. I hope that you find my comments helpful.

Abstract:

- I disagree with the statement that research on choice deferral or refusing alternatives are ignored in classic decision-making research – see Anderson’s (2003) paper on the psychology of doing nothing, and the wealth of related literature on decision avoidance.

Introduction:

- P1 - I see that you cite the Anderson paper here, but you are selective in your two types of decision avoidance – deferral and option refusal. Why do you not discuss further decision avoidance tactics as outlined by Anderson? And the theory behind them? You could also look at some of the work by Power & Alison on decision inertia and redundant deliberation as a form of avoidance e.g., Decision inertia in critical incidents (2018), European Psychologist; Redundant deliberation about negative consequences: decision inertia in emergency responders (2017), Psychology, Public Policy & Law. I just feel you need more background on decision avoidance prior to diving into the assumption that there is little research and only two types – indeed, why did you pick these two types of avoidance to study?

- P1 - Although it can be helpful to outline the structure of a paper, it can also be distracting. Starting your paper with the theoretical background might be more useful and address issues identified in point one.

- P2 – I like the inclusion of the Rubicon model, but it might help to provide a worked example to make it clearer to the reader what distinction is between decision avoidance and action avoidance. It would also be good if you could embed the Rubicon model into other decision avoidance theory (e.g., Anderson) as it reads a bit disjointed at the moment – I’d like to see a richer discussion of theory at the front end.

- P2-3 – Your sections on situational factors/personal factors/quality are informative, but they seem disjointed from your paper. Can you link these sections more clearly to the outstanding research questions that inform your own research?

- P3 – I can’t make sense of your second paragraph about indecisiveness, choice deferral and procrastination – maybe it’s due to a lack of definition but you seem to blur these terms.

- P3 – quality of decision-making. I suggest you read literature from researchers involved in ‘naturalistic decision making’ (NDM) research e.g., Gary Klein, Robert Hoffman, Julie Gore, Nicola Power, Laurence Alison etc. There is a lot of research that rejects the notion of studying right/wrong, good/bad decision-making, and instead focusses on the process of decision-making – being aware that in the real-world it is often impossible to ever know if a decision was truly the ‘best’ (as you never know what would have happened if you’d taken the other option). Reading research by these authors might help to clarify some of the underpinning theory to your research.

- P4 – how do you distinguish between option refusal and choice deferral? Is option refusal not a form of choice deferral (i.e., I refuse options for now). It would be useful to indicate the different between these two processes to make it clear that they are not the same thing.

- P4 – it would be useful to understand more of the theory behind your chosen variables that explains how your predicted effects work e.g., are you just saying that time pressure should reduce indecision, or that this has been found, and if it has been found then why? What’s the psychological explanation?

- P4 – you can’t state that an effect has been ‘proven’ as no research is 100% accurate – you can say there are studies that have evidence to support a given hypothesis, but no study ever ‘proves’ something as a fact.

- P4 – you haven’t made it clear how your tasks are more ‘realistic’ than e.g., lottery tasks. I agree they should be, but before you make this claim it would be useful to have an idea of what you’ve done/will do so I can judge whether I agree your task is more ‘realistic’

- P5 – The debate about rational v intuitive decision-making is interesting, but you seem to have only half discussed it. You also need to consider the role of expertise as well as confidence in decision-making here. Importantly, how does this debate link to your research and what is your stance? This is a big debate in decision science and deserves to be discussed properly if it’s relevant to your study.

- P5 – the NCC section seems odd and out of place. Is something missing here?

- On getting to the end of your introduction I am left wondering what research gap you are filling and/or what makes your study interesting/unique. I’m not saying that there is no gap/interest, but I think you need to make this much clearer. I suggest you restructure the introduction significantly and ensure that your message/point is central to the writing.

Materials and methods

- Participants section should state n of participants.

- “Therefore, every participant received eight variations of one decision situation as a subset of vignettes” – this sentence took a number of readings and I still don’t think I completely understand. Can you rephrase and/or provide an example?

- Table 1 is helpful

- I’m still not sure how you are distinguishing between choice deferral and decision refusal – surely in a simulated vignette design they result in the same outcome (i.e., a choice is not made)? And did you actually make those in the deferral condition choose later? Why would they wait? Did you tell them more information might be given if they defer their choice (often the motivating factor for choice deferral).

Results:

- I’m still not really sure what the difference is between deferral and option refusal (and whether participants perceived them as different options – see above), which makes it difficult for me to interpret your findings. You talk about these as ‘two measures of decision avoidance’ but I’m not convinced because in reality both these options had the same consequences for participants in the study – they didn’t decide. I’m also unsure why you didn’t just collapse them into one category of decision avoidance to compare to those who ‘decided’? You seem to introduce all your findings with ‘decision avoidance’ and then compare the two types, but I don’t see why you’ve done this if you’re going to refer back to the overarching label of ‘decision avoidance’. This might be addressed earlier on in your paper by providing a stronger theoretical background and definitions for your variables.

Discussion:

- It is difficult to comment on your interpretation of findings whilst ambiguity remains about the design of your study.

- In your descriptive overview of decision avoidance behaviour, could you include any data to show whether participants varied in their ‘avoidance’ option choice. I’d be more happy that participants perceived choice deferral and option refusal as distinct and qualitatively different options if they varied in their use of them when avoiding decisions, but if a participants always picked one as the dominant option I’d be worried that participants do not perceive a difference and just pick one of these as their default ‘decision avoidance’ option.

- You say that a strength of the paper is that the decisions are “described in a realistic way”, which I agree would be a strength, but I’m unconvinced that it is true of your study design. Yes you designed your materials to be ‘real-world’ type problems, but there were no consequences for these choices and they were still fictitious, so how does this make them better/more realistic than gambling tasks (for example)? Perhaps it would be better to say you designed them to reflect ‘real world problems’ rather than they are ‘more realistic’. I would interpret realism as meaning the task was immersive and participants felt emotionally invested in choices, but I don’t think that’s what you mean.

Reviewer #2: This paper examines the influence of a number of situational and personal factors on two forms of choice avoidance (option refusal and choice deferral). It reports the results of one experiment in which time pressure, lack of information, and attractiveness of alternatives were manipulated within-subjects, and decision situations (seminar at university vs. plans for evening vs. internship vs. apartment) were manipulated between-subjects. Various personal and control variables were also measured. The impact of all these variables on decision avoidance and decision quality was formally assessed using multi-level models.

My evaluation is based on the PLOS ONE publication criteria, as addressed in the specific questions below:

1. Is the manuscript technically sound, and do the data support the conclusions?

A technically sound manuscript is one that presents well-developed rationales for its hypothesized effects, and tests them in a rigorously conducted experiment with appropriate controls, replications, and sample size. Unfortunately, this manuscript has important shortcomings in both theory and methodology.

Theoretically, the hypotheses are insufficiently motivated. For instance, H1 states that low time pressure implies more decision avoidance than high time pressure. This was justified, in part, based on past research showing the same effect, and in part, based on the vague assertion that the risks of deferral could exceed the benefits the closer a deadline gets. What exactly are the risks and benefits of deferral? and why would the risks of deferral exceed the benefits under high time pressure? The remaining hypotheses are equally underdeveloped, and in the case of H4, appears tautological (chronically indecisive people, by definition, have difficulty making decisions).

Beyond the main effects described in the current hypotheses, the theoretical development would greatly benefit from considering how the various personal and situational factors might interact with each other, instead of assuming a simple additive model.

The conceptual and operational definitions of the main constructs are also problematic. For instance, the manuscript treats option refusal and decision deferral as distinct non-overlapping constructs. Yet choice deferral often implies option refusal. That is, people often defer choice in order to seek other alternatives. Also, the manuscript uses satisfaction with a decision as a proxy measure for decision quality, but decision quality is conceptually very different from satisfaction with the decision. One can be satisfied with a poor-quality decision or dissatisfied with a high-quality decision, especially when asked right after making the decision and before the consequences are known. Many of these problems can be addressed by clearly defining the constructs (e.g., what does decision quality really mean?), and ensuring that operational definitions (i.e., measurement) are consistent with conceptual definitions.

2. Has the statistical analysis been performed appropriately and rigorously?

There are concerns about aspects of the statistical analysis, reporting of the analysis, and interpretation of the findings.

First, not all interdependencies in the data were accounted for. While the analysis accounted for the fact that vignette-level factors were nested within respondents, it failed to account for the nesting within decision situations. The data were instead combined for all four decision situations.

Both the descriptive statistics (under the heading: Decision Avoidance Behavior) and the multi-level models fail to report the effect of decision situation (the between-subjects variable). How do the patterns of decision avoidance vary across the decision situations?

The odds ratios are misinterpreted in several places in the manuscript. For instance, on page 11, referring to an OR of 48.09, it is stated that the probability of decision deferral was 48-times higher with low time pressure than with high time pressure. That is incorrect. The odds are 48-times higher not the probability. Similar statements are found on page 13. Also, OR < 1 are interpreted as very small effects instead of a reduction in the odds. OR < 1 means that the first group is less likely to experience the event than the second group. It doesn’t necessarily imply a very small effect. It would also be useful to report confidence intervals around the ORs.

Table 4: why not present the full table (all the conditions)?

Table 5: the note under the table refers to odds ratios, but there are no odds ratios in the table.

3. Have the authors made all data underlying the findings in their manuscript fully available?

Yes, the data was submitted with the manuscript. However, the analysis script is not available.

4. Is the manuscript presented in an intelligible fashion and written in standard English?

The writing is at times ambiguous, and would benefit from a thorough edit/rewrite.

In addition, in attempting to build a strawman, past research is sometimes misrepresented (e.g., when claiming that the decision-making literature considers choice deferral as dysfunctional, or that past research has not considered a combination of personal and situational factors).

Reviewer #3: This paper presents a vignette-based decision making study focusing on decisional deferral (putting off a decision between options) and decisional avoidance (choosing neither option). As vignette studies go, the methods are generally rigorous, and some of the findings are interesting. The analyses seem to be appropriate and sound, but I am not an expert in the relevant statistical procedures.

I have three primary concerns that significantly undermine my enthusiasm for the paper. First, the paper is very difficult to read. My guess is that thorough editing by a native English speaker would help considerably, but I'm not sure whether simply clarifying the writing will fully resolve the problem. In any case, the writing made it difficult to fully evaluate both the novelty of the study and the precise methods used.

Speaking of novelty, I couldn't quite figure out whether and to what extent the research questions and hypotheses were largely replicating past work or moving into new territory. The introduction cites numerous studies that would seem to address the authors' hypotheses, without making fully clear how the current study departs from or extends beyond that previous work.

Finally, and most problematically, I am simply underwhelmed by studies of decision-making "behavior" that rely on hypothetical decisions based on simple vignettes. I certainly understand that this approach allows each participant to consider scenarios that vary on a number of dimensions; however, this strength isn't very useful if people's decisions fail to reflect decisions they would make in real life. Many decades of social psychological research has confirmed that people are pretty terrible at predicting how they would act in a hypothetical scenario.

As a minor comment, I was surprised that the authors collapsed across their 4 scenario domains without testing whether effects differed across them. The evidence would be somewhat more compelling if the authors instead treated the 4 scenarios as four substudies and then conducted mini-meta-analyses of the effects across those substudies to provide a better sense of how consistent the effects are.

In short, I fear that the methodological shortcomings may be difficult to overcome in a revision. However, due to problems with clarity, it is possible that the authors could convince me of the novelty of their endeavor.

6. PLOS authors have the option to publish the peer review history of their article (what does this mean?). If published, this will include your full peer review and any attached files.

Reviewer #1: Yes: Dr Nicola Power

Reviewer #2: No

Reviewer #3: No

---

## [Author Response · Author response to Decision Letter 0]

12 Mar 2020

Comments of the Editor

1. Thank you for submitting your manuscript to PLOS ONE. After careful consideration, we feel that it has merit but does not fully meet PLOS ONE’s publication criteria as it currently stands. Therefore, we invite you to submit a revised version of the manuscript that addresses the points raised during the review process.

Please find below the reviewers' comments.

We would like to thank the editor for emphasizing the merit of the study and giving us the opportunity to address the concerns.

We did as suggested.

We did as suggested.

3. Additional Editor Comments: I have now collected three reviews from three experts in the field. The reviewers are somehow split, with one recommending rejection and two recommending major revisions. After reading the reviews and the manuscript, I feel that a revision of this manuscript could be publishable. Therefore, I would like to invite you to revise your paper according to the reviewers' comments. Needless to say that all comments must be addressed, especially those from the negative reviewer.

We tried to address all comments in detail.

Comments of Reviewer #1: 

1. Thank you for providing me with the opportunity to review your manuscript. I found it an interesting read and enjoyed reviewing it. I do however have some significant reservations with regards to theory and design, which means I am recommending major revisions. I have two main points. First, I found the theoretical background to the paper was under-developed and confused in places. I wasn’t entirely sure how your paper was unique or what new questions it was addressed. This is not because I believe it lacks these points, I just feel it wasn’t conveyed clearly in the writing. Second, I was unsure of the qualitative difference between choice deferral and option refusal. Did these options actually result in different consequences for participants, or were they both effectively the same in terms of consequences for the participant (i.e., they didn’t make a decision)? I found it difficult to evaluate your findings/discussion as I wasn’t convinced by these options being distinct enough, especially in terms of considering whether participants perceived these 

options as different or the same. I hope that you find my comments helpful.

We would like to thank for this thoughtful evaluation. We are very pleased that you find the manuscript interesting and enjoyed reviewing it. Additionally, the you raised two points, we would like to address and hope to clarify by major revisions in the manuscript. The first point raised addresses the question about what is unique and new about the study. To clarify this, we revised our theoretical background markedly. We agree that lots of research has been done in the area of decision avoidance. Our study focuses on the avoidance of a decision (choice deferral) in separation to other forms former categorized as decision avoidance (e.g., status quo bias, omission bias, inaction inertia), but in our opinion belongs more to action avoidance. To clarify this distinction, we reviewed the paper from Anderson (2003) in more detail and provided the Rubicon Model of Action Phases. Now, we want to clarify the questions, what is unique and what is the qualitative difference between choice deferral and option refusal: 

First, previous studies often lacked the integration of situational as well as personal factors in predicting decision avoidance. We know only two studies that studied the combination of personal indecision and the risk of a decision (Ferrari & Dovidio, 2000; Patalano & Wengrovitz, 2007). In our study, we integrate several situational and personal factors. This is new, because decision avoidance previously has been explored out of the perspective of different but unlinked psychological disciplines (e.g., choice deferral of consumer decisions in marketing research, status quo bias or omission bias as heuristic and biases in cognitive psychology, decision inertia in critical incidents as naturalistic decision-making in ecological psychology, procrastination research in educational or business psychology, chronic indecisiveness in clinical psychology or personality psychology). We agree with Anderson (2003) that there is the need of a more comprehensive view of personal and situational factors as well as advantages and disadvantages across the borders of cognitive, clinical, and social psychology. We tried to clarify our unique contribution throughout the introduction.

Secondly, studies about conditions that affect the form of decision avoidance are still lacking. even though Anderson (2003) had already pointed out this research gap 17 years ago. Therefore, we want to address this research gap by analyzing, how factors previously identified as important influenced different forms of avoidance options. Anderson defines choice deferral as a form of decision avoidance that is characterized by either postponing a decision or refusing to select an option (Anderson, 2003). Previous studies in consumer research often combined both options in a no-choice condition: ‘not chose either of these and look for others’ (e.g., Dhar & Nowlis, 1999; Dhar, 1997, Dhar, 1996). However, in real life these options often don’t go hand in hand. You can refuse options without further searching so that you decide not to buy anything (option refusal without deferring). Or you can search for further options, but still keep in mind the ones you previously saw (deferring without option refusal). In our study we separate these two forms of decision avoidance, so that a decision-maker has the possibility to either choose, defer choice (decision deferral), or refuse choosing any option (option refusal). Decision deferral is defined as to postpone the decision to a later point in time, while option refusal is a decision against existing options. We agree that under some circumstances the two forms of decision avoidance overlap. If a decision-maker refuses current options and searches for further ones, the final decision is also deferred to a later point in time. The options are not completely distinct, but nevertheless, we believed that different factors of the situation and person contribute to the choice of either one or the other. Additionally, under different circumstances the different forms of decision avoidance seem sometimes reasonable, but sometimes dysfunctional as well. In line with this, two forms of decision avoidance were investigated in this study: 1) the option of deferring choice to a later point in time (decision deferral) and 2) the option to not decide at all by refusing both alternatives (option refusal). We had the aims of analyzing 1) which factors of the situation and person influence the choice of either decision deferral or option refusal, and 2) when a decision-maker is satisfied with the choice of an avoidance option? Now, we picked up your point, and elaborate on the two forms in more detail throughout the manuscript.

Abstract:

- I disagree with the statement that research on choice deferral or refusing alternatives are ignored in classic decision-making research – see Anderson’s (2003) paper on the psychology of doing nothing, and the wealth of related literature on decision avoidance.

Thank you for this comment. We agree about the wealth of literature on decision avoidance. Our statement intended to state that “classic” decision making research traditionally ignores the option of choice deferral or classifies choice deferral as dysfunctional. Furthermore, previous studies did not differentiate between choice deferral and refusing alternatives as different decision avoidance options. Nevertheless, we reformulate the paragraph to clarify our point: 

‘The avoidance of a decision is a phenomenon that has been studied in various forms and psychological disciplines. Nevertheless, previous studies often lacked the integration of situational as well as personal factors in predicting decision avoidance. Additionally, studies about conditions that affect the form of decision avoidance are still lacking. Therefore, this study investigated how situational and personal factors influenced two different forms of decision avoidance: (1) the option of deferring choice to a later point in time (decision deferral) and (2) the option to not decide at all by refusing both alternatives (option refusal).’ (p.2, line 9-18)

Introduction:

- P1 - I see that you cite the Anderson paper here, but you are selective in your two types of decision avoidance – deferral and option refusal. Why do you not discuss further decision avoidance tactics as outlined by Anderson? And the theory behind them? You could also look at some of the work by Power & Alison on decision inertia and redundant deliberation as a form of avoidance e.g., Decision inertia in critical incidents (2018), European Psychologist; Redundant deliberation about negative consequences: decision inertia in emergency responders (2017), Psychology, Public Policy & Law. I just feel you need more background on decision avoidance prior to diving into the assumption that there is little research and only two types – indeed, why did you pick these two types of avoidance to study?

Thank you very much for pointing out the interesting literature on decision inertia in critical incidents to us that we read with pleasure and added to our manuscript. We agree that research on decision avoidance is a wide field. Since we did not prepare a review, our paper is partly selective in referring to the forms of decision avoidance we focused our study on. Nevertheless, we recognized the need to draw a wider bow and to better explain, why we picked these two types of avoidance to study.

We firstly added an overview of background on decision avoidance at the beginning of the introduction:

‘Decision avoidance is a phenomenon that has been studied in various forms and psychological disciplines (Anderson, 2003): choice deferral of consumer decisions in marketing research (e.g., Dhar, 1996), heuristic and biases of decision avoidance in cognitive psychology (e.g., Tversky & Shafir, 1992), decision inertia in critical incidents in naturalistic decision-making research (e.g., Power & Alison, 2018), procrastination research in educational or business psychology (e.g., Solomon & Rothblum, 1984), chronic indecisiveness in clinical psychology or personality psychology (e.g., Leykin & DeRubeis, 2010). Altogether, there is a bunch of studies in the context of decision avoidance, but they have explored the phenomenon out of the perspective of different and partly unlinked psychological disciplines. In line with Anderson (2003), the present study aimed to complement different research traditions to a more comprehensive view. Therefore, this study wants to combine situational and personal predictors of decision avoidance and analyzes how participants experience their avoidance behavior.’ (p. 3, line 45-56) 

We also added the following paragraph to address the avoidance tactics outlined by Anderson:

‘Anderson describes four different forms of decision avoidance: People often like to stay with the familiar (status quo bias, Samuelson & Zeckhauser, 1988), they prefer omissions over actions in risky decision situations (omission bias, Ritov & Baron, 1990), omit action when a similar more attractive option was recently missed (inaction inertia, Tykocinski et al., 1995) or defer decisions to look for further options or information (choice deferral). Studies on status quo bias and omission bias formulate hypothetical situation in which the decision-maker can do something (e.g., vaccinate or investing stocks) or omit to do it and staying with the status quo, especially in a high-risk situation with potentially harmful consequences (Ritov & Baron, 1990; Samulson & Zeckhausen, 1988). In there, a balancing of potential risks and the fear of negative consequences play an important role in the decision process. These are other basic conditions as in studies on choice deferral in consumer decisions, where hypothetical situations are formulated in which the decision-maker can decide on various products (e.g., buy camera A or B) or has the opportunity to search for further options. In there, a balancing of preferences goes along with the option not to choose at all. These decisional situations should be less stressful unless specific situational circumstances increase the pressure (e.g., time pressure), uncertainty (e.g., lack of information) or discomfort (e.g., unattractive options) or if personal characteristics impair confident and rational decision making (e.g., chronic indecisiveness, anxiety).‘ (p. 3-4, line 58-83)

We added to following paragraph to clarify, why we picked the two types of avoidance to study:

‘In real life, decisions are often deferred to a later point in time to collect more information (Corbin, 1980) or options are refused, because no alternative satisfies (Beach, 1993). Anderson defines choice deferral as a form of decision avoidance that is characterized by either postponing a decision or refusing to select an option (Anderson, 2003). Previous studies in consumer research often combined both options in a no-choice condition (not chose either of these and look for others; e.g., Dhar & Nowlis, 1999; Dhar, 1997, Dhar, 1996). Studies about conditions that affect different forms of decision avoidance are still lacking, even though Anderson (2003) had already pointed out this research gap 17 years ago. Therefore, we want to address this research gap by analyzing how situational and personal factors influence two different forms of the avoidance option (decision deferral and option refusal). Hence, in our study, we separate these two forms of decision avoidance, so that a decision-maker has the possibility to either choose, defer choice (decision deferral), or refuse to choose any option (option refusal). Decision deferral is defined as to postpone the decision to a later point in time, while option refusal is a decision against existing options. In line with this, two forms of decision avoidance were investigated in this study: (1) the option of deferring choice to a later point in time (decision deferral) and (2) the option to not decide at all by refusing both alternatives (option refusal).’ (p. 4-5, line 103-136)

Furthermore, we referred to the added literature within the hypotheses: ‘Therefore, in decision situations of high stress, a decision between two unattractive options is often avoided (as studied in status quo bias and omission bias, Ritov & Baron, 1990; Samulson & Zeckhausen, 1988). Additionally, research about decisions in the context of critical incidents showed that redundant deliberation about negative consequences results in behavioral inaction and even worse outcomes (decision inertia, Power & Alison, 2018).’ (p.11, line 470-474)

- P1 - Although it can be helpful to outline the structure of a paper, it can also be distracting. Starting your paper with the theoretical background might be more useful and address issues identified in point one.

Thank you for this notice. We now start with a more general introduction in the theoretical background and restructured the sections. After an overview of previous literature on decision avoidance from various psychological disciplines, we define different forms on decision avoidance and classify them in a process model of Action. Afterwards, we introduce previous literature on situational as well as personal predictors of decision avoidance as well as measurements of decision-making quality. The introduction closes with the overall aim of the study, research questions and hypotheses. 

- P2 – I like the inclusion of the Rubicon model, but it might help to provide a worked example to make it clearer to the reader what distinction is between decision avoidance and action avoidance. It would also be good if you could embed the Rubicon model into other decision avoidance theory (e.g., Anderson) as it reads a bit disjointed at the moment – I’d like to see a richer discussion of theory at the front end.

Thanks for this comment. We appreciate the idea of providing an example. Therefore we added the following to better distinguish between decision avoidance and action avoidance as well as differentiate between choice deferral and option refusal:

‘For example, one can decide whether to take Course A or Course B in the gym (predecisional phase). One can delay this decision for a longer time, e.g., because one is still thinking about what to like better (decision deferral). One can also simply decide against both options and do not take any course (option refusal). Then the decision process is either finished, or one looks for new alternatives (predecisional phase starts again). Assuming a decision for a course has been made, and one has registered (crossing the Rubicon). Then the decision process is finished, and the preactional phase begins. In there, one can postpone going to the decided course, e.g., because it is difficult to motivate oneself (action avoidance).’ (p. 4, line 94-101)

We link the Rubicon model now to the forms of decision avoidances outlined by Anderson:

‘This study will focus on what Anderson (2003) has described as choice deferral to mainly focus on the avoidance of choice as distinct from the avoidance of action. Previous studies often blur terms of decision avoidance by not separating between action avoidance and decision avoidance.’ (p. 4, line 84-87)

Furthermore, we pick up the Rubicon model in sections about personal factors later on:

‘To refer to the Rubicon Model of Action Phases, in this study, the focus is on decision avoidance and not on action avoidance. Furthermore, this study aims to analyze the conditions of the choice of an avoidance option without predefining deferral as dysfunctional procrastination. Therefore, we consider the factor of personal indecisiveness as a general pattern.’ (p. 7-8, line 308-329)

And: ‘Referring to the Rubicon Model of Action Phases, the predecisional phase is characterized by broad information processing (Gollwitzer, 1990), for which a low NCC is considered to be a crucial part. On the other hand, the postdecisional phase already focuses on implementation, which requires focusing rather than to broaden the view, and therefore a high NCC is important (Kruglanski & Webster, 1996).’ (p. 8, line 334-338)

- P2-3 – Your sections on situational factors/personal factors/quality are informative, but they seem disjointed from your paper. Can you link these sections more clearly to the outstanding research questions that inform your own research?

Thank you for this comment. We revised the sections markedly to refer to our primary aims and specific methodology. Additionally, we added the following outstanding research questions:

‘Overall, many situational factors have been shown to influence decision avoidance, but no previous study has compared the different effects on decision deferral and option refusal as two different forms of decision avoidance.’ (p. 6, line 248-250)

‘In this study, we integrate both perspectives of personal and situational factors in predicting decision avoidance. As situational factors could be varied in different vignettes, personal factors will be captured by questionnaires. Furthermore, this study also had the aim of investigating the experience of decision avoidance in daily decision situations to combine rational and emotional factors of decision avoidance in line with Anderson (2003).‘ (p. 8, line 339-343)

‘Overall, this study aimed 1) to investigate the impact of personal and situational factors that influence decision deferral and option refusal as two forms of decision avoidance and 2) to evaluate, the experienced satisfaction of either choosing, deferring, or refusing depending on manipulated context conditions.’ (p- 9, line 389-392)

- P3 – I can’t make sense of your second paragraph about indecisiveness, choice deferral and procrastination – maybe it’s due to a lack of definition but you seem to blur these terms.

We apologize if the different terms are confusing. This is mainly due to different usage of terms in different disciplines. However, we tried to clarify this by providing additional definition of procrastination and theoretical grounding: 

‘Indecisiveness as a personality variable is described as the characteristic of having chronic difficulties with decisions and to defer those as a result (Crites, 1969). In this definition, the subjective experience of difficulty and the decision-making behavior are linked. Indecisiveness is associated with psychopathological disorders (Radford, Mann, & Kalucy, 1986). Therefore, studies within clinical or educational psychology do often classify choice deferral as a subdomain of procrastination. Procrastination is conceptualized as a personal disposition to postpone, delay, and thereby avoid actions or decisions that have to be taken (Milgram & Tenne, 2000). In general, procrastination is characterized as a maladaptive pattern (Ferrari, Johnson, & McCown, 1995) that contradicts the personal aims by definition (Sabini & Silver, 1982; Schouwenburg & Lay, 1995). As a differentiation, decisional procrastination has been linked to neuroticism and self-regulation, while task avoidance procrastination is linked to lower conscientiousness and self-control (Milgram & Tenne, 2000). To refer to the Rubicon Model of Action Phases, the focus of this study is on decision avoidance and not on action avoidance. Furthermore, this study aims to analyze the conditions of the choice of an avoidance option without predefining deferral as dysfunctional procrastination. Therefore, we consider the factor of personal indecisiveness as a general pattern. ‘ (p. 7-8, line 298-329)

- P3 – quality of decision-making. I suggest you read literature from researchers involved in ‘naturalistic decision making’ (NDM) research e.g., Gary Klein, Robert Hoffman, Julie Gore, Nicola Power, Laurence Alison etc. There is a lot of research that rejects the notion of studying right/wrong, good/bad decision-making, and instead focusses on the process of decision-making – being aware that in the real-world it is often impossible to ever know if a decision was truly the ‘best’ (as you never know what would have happened if you’d taken the other option). Reading research by these authors might help to clarify some of the underpinning theory to your research.

Thank you for this comment. We read the literature from researchers involved in ‘naturalistic decision-making’. We agree that this was not explicitly referred to in our first version. Now, we enriched the section with a reference by Lipshitz, Klein, Oranasu and Salas 2001: ‘Therefore, studies in naturalistic decision making are process-oriented without relying on normative choice models (Lipshitz, Klein, Oranasu et al., 2001). The study aimed to analyze how participants experience their avoidance behavior to capture functional and dysfunctional avoidance out of a subjective perspective instead of providing predefined best options. Thus, normative measurements to assess the quality of decision-making behavior (like the multi-attribute utility theory; Baron, 2004) are not applicable in this context.’ (p. 8, line 348-353)

Furthermore, we included a whole paragraph about the topic in the discussion (see later).

- P4 – how do you distinguish between option refusal and choice deferral? Is option refusal not a form of choice deferral (i.e., I refuse options for now). It would be useful to indicate the different between these two processes to make it clear that they are not the same thing.

Decision deferral is defined as to postpone the decision to a later point in time, while option refusal is a decision against existing options. We agree that under some circumstances the two forms of decision avoidance overlap. If a decision-maker refuses current options and searches for further ones, the final decision is also deferred to a later point in time. The options are not completely distinct, but nevertheless, we believed that different factors of the situation and person contribute to the choice of either one or the other. Additionally, under different circumstances other forms of decision avoidance seem reasonable or dysfunctional avoidance.

We tried to clarify the differences of the two forms by giving examples in the Rubicon Model (see previous comment) and providing this paragraph:

‘In real life, decisions are often deferred to a later point in time to collect more information (Corbin, 1980) or options are refused, because no alternative satisfies (Beach, 1993). Anderson defines choice deferral as a form of decision avoidance that is characterized by either postponing a decision or refusing to select an option (Anderson, 2003). Previous studies in consumer research often combined both options in a no-choice condition (not chose either of these and look for others) (e.g., Dhar & Nowlis, 1999; Dhar, 1997, Dhar, 1996). Studies about conditions that effect different form of decision avoidance are still lacking, even though Anderson (2003) had already pointed out this research gap 17 years ago. Therefore, we want to address this research gap by analyzing, how situational and personal factors influence two different forms of the avoidance option (decision deferral and option refusal). Hence, in our study we separate these two forms of decision avoidance, so that a decision-maker has the possibility to choose, to defer choice (decision deferral), and to refuse choosing any option (option refusal). Decision deferral is defined as to postpone the decision to a later point in time, while option refusal is a decision against existing options. In line with this, two forms of decision avoidance were investigated in this study: 1) the option of deferring choice to a later point in time (decision deferral) and 2) the option to not decide at all by refusing both alternatives (option refusal). ‘ (p.4, line 103-136)

We added an additional paragraph under the section ‘Subjective Decision-Making Quality: Post-Decisional Satisfaction’:

‘Depending on personal and situational conditions, both strategies of decision deferral and option refusal could be either adaptive or dysfunctional indecisiveness: To defer a decision for further deliberation could be a reasonable step, e.g., to collect more information if there are enough recourses and less time pressure. On the other hand, extensive rumination, e.g., because of the fear of a wrong decision or negative consequences could be dysfunctional avoidance, if no information gain is expected. To refuse current options could be rational, if previous options are unattractive and one is not dependent on taking an option, e.g., staying at home and neither go to party A nor to party B, if both do not satisfy. However, refusing options can also be inappropriate or worsen the initial situation, e.g., if one refuses quite good job offers, despite one has financial problems. In this study, we assume that the decision-maker would be more satisfied with the choice of an avoidance option if he or she judges the avoidance as rational rather than dysfunctional avoidance. Therefore, in this work, satisfaction with the decision-making behavior was used as a subjective indicator for the quality of decision-making behavior.’ (p.9. line 374-387)

Furthermore, we added to the hypotheses section specific considerations about how the factors are expected to impact decision deferral and option refusal differently:

H1: ‘Our study seeks to replicate these findings in further, complementing decision situations and tests for the first time whether the effects differ between decision deferral and option refusal. In there, we expect the effect to be greater on decision deferral than on option refusal, because in the context of high time pressure decision deferral implies no gain, but option refusal could still be reasonable. (p. 10, line 429-433) 

‘H1: Low time pressure implies more decision avoidance (especially decision deferral) than high time pressure.’ (p. 10, line 434-435) 

H2: ‘Furthermore, this study firstly investigates whether the effects differ between decision deferral and option refusal. Due to the higher uncertainty, which can be concluded with more time, decision deferral seems more evident than option refusal under the condition of a high lack of information. Nevertheless, it is possible that if both options are associated with an as yet unclear risk, they are more likely to be rejected.’ (p. 10, line 434-435) 

‘H2: Lack of information implies more decision avoidance (especially decision deferral) than full information.’ (p. 11, line 466-467)

‘Therefore, in the context of consumer decisions, option refusal increases if alternatives are not attractive (25). In there, the avoidance seems like an adequate strategy and no dysfunctional avoidance. Therefore, this study investigates how the attractiveness of the choice-set influences the choice of decision deferral or option refusal in the context of daily decision situations. In there, larger effects are expected on option refusal than on decision deferral.’ (p. 11, line 476-481)

‘H3: Decisions within an unattractive choice-set imply more decision avoidance (especially option refusal) than decisions within an attractive option-set.’ (p. 11-12, line 482-490)

H4: ‘Patalano and Wengrovitz reported that indecisive participants did not show uniformly increased deferral, but rather that their deferral behavior is less strongly adapted to context conditions like increased risk. Therefore, this study wants to show if the indecisiveness of a person has an additional predictive effect compared to the situational factors in the prediction a decision avoidance behavior. Therefore, it is included as the main personal factor in the analysis models. It has never been studied before, if chronic indecisiveness impacts decision deferral and options refusal as two options of decision avoidance differently. Since indecision is essentially associated with more time to decide, we expect it to be mainly associated with decision deferral and less with option refusal.’ (p. 12, line 496-504) 

‘H4: The higher the indecisiveness of a person, the more decision avoidance is shown (especially decision deferral).’ (p. 12, line 505-506)

- P4 – it would be useful to understand more of the theory behind your chosen variables that explains how your predicted effects work e.g., are you just saying that time pressure should reduce indecision, or that this has been found, and if it has been found then why? What’s the psychological explanation?

Thank you for this comment. We enriched the psychological explanation in this section:

‘However, higher time pressure should reduce decision deferral, because the costs of deferral (e.g., risk that previous options change or that you lose options) could exceed possible benefits (e.g., the gain of information or clarity) the closer a deadline gets. Under a condition of low time pressure, decision deferral could be reasonable to search for further information or alternatives or to increase deliberation about consequences and to capture the complexity (Milgram & Tenne, 2000). Besides the expected gain of information, deferral has its added value by increasing certainty with decision and acceptance of unpleasant decisions (Eliaz & Schotter, 2010; Tykocinski & Ruffle, 2003). The relation between the length of time and the deferral and avoidance behavior has already been shown (Dhar & Nowlis, 1999; Tversky & Shafir, 1992; Tykocinski & Ruffle, 2003). Our study seeks to replicate these findings in further, complementing decision situations and tests for the first time whether the effects differ between decision deferral and option refusal. In there, we expect the effect to be higher on decision deferral than on option refusal, because in the context of high time pressure decision deferral implies no gain, but option refusal could still be reasonable.’ (p. 10, line 421-433)

- P4 – you can’t state that an effect has been ‘proven’ as no research is 100% accurate – you can say there are studies that have evidence to support a given hypothesis, but no study ever ‘proves’ something as a fact.

We totally agree with your comment and changed our wording accordingly: ‘The relation between the length of time and the deferral and avoidance behavior has already been shown (Dhar & Nowlis, 1999; Tversky & Shafir, 1992; Tykocinski & Ruffle, 2003).’ (p. 10, line 428-429)

- P4 – you haven’t made it clear how your tasks are more ‘realistic’ than e.g., lottery tasks. I agree they should be, but before you make this claim it would be useful to have an idea of what you’ve done/will do so I can judge whether I agree your task is more ‘realistic’

We included a paragraph about the decision situations within the introduction now:

‘In this study, therefore, different vignettes are designed to explore the impact of situational conditions on decision deferral and option refusal by varying them systematically. In there, decision situations were developed to represent common and meaningful situations in a student’s everyday life. Previous studies primarily focused on the context of product purchases (Dhar, 1996, 1997a, 1997b; Dhar & Nowlis, 1999; Luce, 1998), but also other daily decision situations like Friday night activity (Dhar, 1997a), seminar (Ferrari & Dovidio, 2000; Patalano & Wengrovitz, 2007) or apartment (Dhar & Nowlis, 1999; Dijksterhuis, 2004; Tversky & Shafir, 1992) have been studied. This study attaches importance to the fact that vignettes reflect familiar decisional situations that are formulated in a realistic way and fit the sample of student participants. Therefore, this study uses no ecological approach (like in field studies) but realistically simulated environments (like, e.g. Smith, Giffin, Rockwell, & Thomas, 1986) to combine concerns of naturalistic decision-making as well as controlled experimental scenarios (Todd & Gigerenzer, 2001).’ (p. 6, line 251-260)

- P5 – The debate about rational v intuitive decision-making is interesting, but you seem to have only half discussed it. You also need to consider the role of expertise as well as confidence in decision-making here. Importantly, how does this debate link to your research and what is your stance? This is a big debate in decision science and deserves to be discussed properly if it’s relevant to your study.

Thank you for this comment. We agree that there is a big and important debate about rational vs. intuitive decision-making in decision science. Additionally, the role of expertise as well as confidence in decision-making are important factors in decision-making research. In our study, we analyzed daily decision situations within a student’s life, why no particular expert knowledge was required for our student sample. However, we agree that confidence in decision making is a relevant factor. We have grasped both the difficulty of the decision and the personal self-confidence as decision-maker. Additionally, we included subscales to capture rational and intuitive decision-making styles as relevant personal factors and analyzed the association with decision deferral and option refusal. In our study, the confidence was negatively associated with decision deferral, the intuitive style was positively associated with option refusal and the rational style showed no correlation with either decision deferral nor option refusal. Overall, these results were interesting and picked up in the discussion section in more detail. Since the scope of our study is limited, we only picked out central points from the debate that are relevant for decision avoidance and left out aspects of rational vs. intuitive decision-making in general. Nevertheless, we now tried to enriched the introduction section with a reference to dual-process theory and now better refer to our expected associations with decision deferral in the hypotheses section:

‘In literature, different classifications of decision-making styles are proposed: Maybe the most classical distinction is the one between rational and intuitive decision-making (dual-process theory; Evans, 2008).’ (p. 7, line 283-284)

‘The rational style requires more time to make decisions compared to the intuitive style (Scott & Bruce, 1995). Therefore, rational decision making could be associated with decision deferral because of a further search for information or deliberation. However, this association is also questionable because the rational style is often classified as adaptive and is contrasted to avoiding choice (Mann et al., 1997). On the other hand, the intuitive style has its strength in automatic and fast decision-making (Glöckner & Betsch, 2008), which is why this style is associated with a decreased deferral. Confidence in decision-making should decrease avoidance behavior (especially decision deferral), while anxiety should increase avoidance behavior according to the conflict theory (Janis & Mann, 1977).’ (p.12-13, line 510-525)

- P5 – the NCC section seems odd and out of place. Is something missing here?

We better integrated the NCC section into the manuscript by referring to previous theory about the Rubicon model and decision avoidance:

‘Referring to the Rubicon Model of Action Phases, 1990), for which a low NCC is considered to be a crucial part. On the other hand, the postdecisional phase already focuses on implementation, which requires focusing rather than to broaden the view, and therefore a high NCC is important (Kruglanski & Webster, 1996). ‘ (p.8, line 334-338)

Furthermore, we added some considerations about the association of NCC with decision avoidance within the hypotheses section: ‘It could be expected that the NCC influences the two forms of decision avoidance differently: A low NCC is associated with a broadening of the decision process and should, therefore, increase decision deferral. On the other hand, option refusal is an avoidance option that possibly closes the decision process, why it should be associated with a high NCC. In conclusion, NCC is additionally included as a personal variable within the correlational analyses to explore the association with the two different forms of decision avoidance.‘ (p.13, line 535-541)

- On getting to the end of your introduction I am left wondering what research gap you are filling and/or what makes your study interesting/unique. I’m not saying that there is no gap/interest, but I think you need to make this much clearer. I suggest you restructure the introduction significantly and ensure that your message/point is central to the writing.

We restructured the introduction markedly to better emphasize our research idea that fills a gap within previous literature. Furthermore, we added a paragraph at the end of the instruction to summarize central aims and hypotheses:

‘Summarizing, this study aimed to investigate the impact of personal and situational factors that influence decision deferral and option refusal as two forms of decision avoidance. To achieve this, the study tests the hypotheses, if decision avoidance is increased under low time pressure, high lack of information, low attractiveness of the choice set, and high personal indecisiveness. In there, it will be explored, if predictors differ between the two forms of decisional deferral and option refusal. Decision deferral should be increased by low time pressure, high lack of information, and high personal indecisiveness. Option refusal should especially be increased by the low attractiveness of the choice set. Furthermore, the experienced satisfaction of either choosing, deferring, or refusing is explored depending on manipulated context conditions. The satisfaction serves as a subjective marker for the quality of the decision-making behavior.’ (p.14-15, line 582-600)

Furthermore, we provided more clarity to the hypotheses section by adding our unique research idea that fill an important gap in previous literature (as seen above): 

H1: ‘Our study seeks to replicate these findings in further, complementing decision situations and tests for the first time whether the effects differ between decision deferral and option refusal. In there, we expect the effect to be greater on decision deferral than on option refusal, because in the context of high time pressure decision deferral implies no gain, but option refusal could still be reasonable.’ (p. 10, line 429-433) 

H2: ‘Furthermore, this study firstly investigates whether the effects differ between decision deferral and option refusal. Due to the higher uncertainty, which can be concluded with more time, decision deferral seems more evident than option refusal under the condition of a high lack of information. Nevertheless, it is possible that if both options are associated with an as yet unclear risk, they are more likely to be rejected.’ (p. 11, line 476-481)

H4: ‘It has never been studied before, if chronic indecisiveness impacts decision deferral and options refusal as two options of decision avoidance differently. Since indecision is essentially associated with more time to decide, we expect it to be mainly associated with decision deferral and less with option refusal.’ (p. 12, line 496-504)

We added information about our expectancies about the connection with decision deferral and option refusal to the section of our control variables as well:

‘Therefore, a positive association between importance and decision deferral is expected from previous studies. However, this need not apply to option refusal as the inverse relationship can be assumed: If a decision is important, the costs of refusing options could be higher as in less important decisions.’ (p.14, line 565-568)

‘It is expected that the similarity influences decision deferral as well as option refusal, but previous studies analyzed them separately. Similarity should increase decision deferral because more deliberation is needed to grasp subtle differences, but high similarity could increase option refusal as well if a decision-maker seeks for the one option that outshines others.’ (p.14, line 573-577)

Materials and methods

- Participants section should state n of participants.

We did as suggested and added the following sentence: ‘Overall, n=312 participants were analyzed within the study.’ (p.15, line 607-608)

The detailed study flowchart is presented in the results section.

- “Therefore, every participant received eight variations of one decision situation as a subset of vignettes” – this sentence took a number of readings and I still don’t think I completely understand. Can you rephrase and/or provide an example?

We tried to clarify the sentence and provided an example and referred to Table 1: Therefore, every participant received a subset of the vignettes that contained eight variations of one decision situation (e.g., all variations of the decision situation seminar at university, see the first line in Table 1).’ ’ (p.15-16, line 619-627)

- Table 1 is helpful

Thank you.

- I’m still not sure how you are distinguishing between choice deferral and decision refusal – surely in a simulated vignette design they result in the same outcome (i.e., a choice is not made)? And did you actually make those in the deferral condition choose later? Why would they wait? Did you tell them more information might be given if they defer their choice (often the motivating factor for choice deferral).

We would like to clarify your questions. The participants were only given the introduction to behave as in real life and that they could either choose, defer the choice to a later point in time or refuse to choose any option (see p. 20, line 701-702). We did not make those in the deferral condition choose later nor did we give them any additional information. We agree that in this simulated vignette design, participants do not have real consequences or certain reasons to wait. We know that in other studies that focus on the process character of decisions, participants could get further information (e.g., Ferrari & Dovidio, 2000). This is also an exciting approach. However, we believe that this is also very artificial because participants do not have to do anything to get the information. Besides, in real life there are many reasons to wait (e.g., to gain clarity, asking a friend, other priorities etc.). We deliberately wanted to keep the reasons for deferral or option refusal open. Therefore, we did not set any incentives in the experimental situation, but encouraged the participants to put themselves in it as if it were a real decision.

Results:

- I’m still not really sure what the difference is between deferral and option refusal (and whether participants perceived them as different options – see above), which makes it difficult for me to interpret your findings. You talk about these as ‘two measures of decision avoidance’ but I’m not convinced because in reality both these options had the same consequences for participants in the study – they didn’t decide. I’m also unsure why you didn’t just collapse them into one category of decision avoidance to compare to those who ‘decided’? You seem to introduce all your findings with ‘decision avoidance’ and then compare the two types, but I don’t see why you’ve done this if you’re going to refer back to the overarching label of ‘decision avoidance’. This might be addressed earlier on in your paper by providing a stronger theoretical background and definitions for your variables.

We tried to address these points by providing a stronger theoretical background and definitions of our two forms of avoidance in the introduction. Previous literature mainly studied decision avoidance in one category, but the factors have never been explored separately. Therefore, the strategy to just collapse decision deferral and option refusal into one category of decision avoidance would mainly replicate previous findings for other situations. This contradicts our primary research interest, which is to show that the two forms of decision avoidance are differently affected by factors of the situation and the person. To clarify this, we now distinguished specific considerations within the hypotheses section (see also comments above). 

Discussion:

- It is difficult to comment on your interpretation of findings whilst ambiguity remains about the design of your study.

We hope that our markedly revisions provide clarity about the design of our study.

- In your descriptive overview of decision avoidance behaviour, could you include any data to show whether participants varied in their ‘avoidance’ option choice. I’d be more happy that participants perceived choice deferral and option refusal as distinct and qualitatively different options if they varied in their use of them when avoiding decisions, but if a participants always picked one as the dominant option I’d be worried that participants do not perceive a difference and just pick one of these as their default ‘decision avoidance’ option.

Thank you very much for your interesting idea. We now added data about how participants varied in their choice of an avoidance option: n=52 (16.7%) never used an avoidance option, n=56 (17.9%) only used the option of ‘decision deferral’ at least once, n=109 (34.9%) only used the option of ‘option refusal’ at least once, and n=95 (30.4%) used both decision deferral and option refusal depending on the vignette. This data shows that option refusal is more popular than decision deferral and there are individual differences in the preference for a specific avoidance option. Our study showed that individual indecisiveness goes along with a preference for ‘decision deferral’. However, about one third of the participants varied in their use of an avoidance option depending on the situational conditions of the vignettes. This supports our conceptualization that participants perceive the two options as distinct and qualitatively different. 

We added the following to the results section: ‘Overall, 16.7% of the participants never used an option of decision avoidance, 17.9% only used the option to defer a decision at least once, 34.9% only used option refusal at least once, and 30.4% used both decision deferral and option refusal depending on the vignette.’ (p. 22-23, line 784-793)

We included into the discussion: ‘Finally, decision deferral and option refusal are different forms of decision avoidance but not necessarily have to be distinct. However, our results showed that participants seem to experience the two forms as qualitatively different because there were clear intraindividual variations in the use of the form of avoidance option depending on situational variations in the vignettes.’ (p.37, line 1328-1332)

- You say that a strength of the paper is that the decisions are “described in a realistic way”, which I agree would be a strength, but I’m unconvinced that it is true of your study design. Yes you designed your materials to be ‘real-world’ type problems, but there were no consequences for these choices and they were still fictitious, so how does this make them better/more realistic than gambling tasks (for example)? Perhaps it would be better to say you designed them to reflect ‘real world problems’ rather than they are ‘more realistic’. I would interpret realism as meaning the task was immersive and participants felt emotionally invested in choices, but I don’t think that’s what you mean.

Thanks for the comments. We agree that we claim our decision tasks to be ‘more realistic’, because they are ‘described in a realistic way’ and they reflect ‘real world problems’ as they are conceptualized as daily decision situations in a student’s life. To clarify this, we therefore added the following description within the introduction: ‘In this study, therefore, different vignettes are designed to explore the impact of situational conditions on decision deferral and option refusal by varying them systematically. In there, decision situations were developed to represent common and meaningful situations in a student’s everyday life. Previous studies primarily focused on the context of product purchases (Dhar, 1996, 1997a, 1997b; Dhar & Nowlis, 1999; Luce, 1998), but also other daily decision situations like Friday night activity (Dhar, 1997a), seminar (Ferrari & Dovidio, 2000; Patalano & Wengrovitz, 2007) or apartment (Dhar & Nowlis, 1999; Dijksterhuis, 2004; Tversky & Shafir, 1992) have been studied. This study attaches importance to the fact that vignettes reflect familiar decisional situations that are formulated in a realistic way and fit the sample of student participants. Therefore, this study uses no ecological approach (like in field studies) but realistically simulated environments (like, e.g. Smith, Giffin, Rockwell, & Thomas, 1986) to combine concerns of naturalistic decision-making as well as controlled experimental scenarios (Todd & Gigerenzer, 2001).’ (p.6, line 251-260)

To ensure external validity, we have assessed how realistically participants experienced the vignettes. The results showed that participants rated the decision situations to be rather realistic and that they are quite familiar with similar situations. Besides, decision situations were clearly not considered trivial as they were rated as important and moderately difficult. We now added this data within the manuscript to support the validity of our efforts. 

In the method section: ‘To ensure external validity the following was captured: Participants were asked as to how realistically they experienced the decision situation and if they had their experiences with a similar decision situation in the past (one-item scales of 0-100).’ (p.20, line 714-717)

In the results section: ‘The participants rated the decision situations to be rather realistic (M = 70.13, SD = 22.28; scale 0-100). Some of the participants had already had their own experiences with a similar situation in the past (M = 45.82, SD = 30.88; scale 0-100). On average, decision situations were assessed as rather important (M = 70.60, SD = 22.76; scale 0-100) and moderately difficult (M = 42.00, SD = 21.18; scale 0-100).’ (p.22, line 764-769)

Furthermore, we included a whole paragraph about realistic decision-making scenarios in the discussion: ‘This study takes up some concerns that naturalistic decision-making research has about classic decision-making research (e.g., Lipshitz, Klein, Orasanu et al., 2001): The vignettes reflect familiar decisional situations that are formulated in a realistic way and fit our sample of student participants. Additionally, the study focuses on process-orientation in contrast to input-output-orientation by analyzing when people chose an avoidance option and how satisfied they are with it instead of providing predefined best choices. Nevertheless, this study evaluated hypothetical decision situations and no real-life behavior to systematically vary different dimensions to evaluate the importance of the prediction of an avoidance option. With this approach, we are in line with Todd and Gigerenzer (2001) that argue no general contradiction between formal modeling and descriptively valid and task-specific real-world decisions. Therefore, we consider important real-life context conditions like time pressure and incompleteness of the information but formalize them in standardized vignettes to combine different advantages.’ (p. 36, line 1298-1309)

Nevertheless, we agree that hypothetical scenarios provide some risks. Therefore, we already had stated within the limitation section: ‘The present study was conducted with self-developed vignettes, so that results could deviate from real decision-making behavior (Groß & Börensen, 2009). It can be assumed that decision avoidance is of greater importance in real life decision situations and the possibility of socially desirable responses has to be taken into account, because decision deferral is considered as a weakness. Nevertheless, vignette studies provide high internal validity and allow to hierarchically assess personal and situational factors (Atzmüller & Steiner, 2010).’ (p.36-37, line 1310-1320)

However, we are still convinced of the advantages as we state ‘Finally, decision situations were described in a realistic way and stimulus material and student sample constituted a good fit. This was also supported by our participants that assessed the decision situations to be rather realistic. Overall, this strengthens external validity.’ (p. 37, line 1341-1344) 

Comments of Reviewer #2: 

This paper examines the influence of a number of situational and personal factors on two forms of choice avoidance (option refusal and choice deferral). It reports the results of one experiment in which time pressure, lack of information, and attractiveness of alternatives were manipulated within-subjects, and decision situations (seminar at university vs. plans for evening vs. internship vs. apartment) were manipulated between-subjects. Various personal and control variables were also measured. The impact of all these variables on decision avoidance and decision quality was formally assessed using multi-level models.

My evaluation is based on the PLOS ONE publication criteria, as addressed in the specific questions below:

1. Is the manuscript technically sound, and do the data support the conclusions?

A technically sound manuscript is one that presents well-developed rationales for its hypothesized effects, and tests them in a rigorously conducted experiment with appropriate controls, replications, and sample size. Unfortunately, this manuscript has important shortcomings in both theory and methodology.

Theoretically, the hypotheses are insufficiently motivated. For instance, H1 states that low time pressure implies more decision avoidance than high time pressure. This was justified, in part, based on past research showing the same effect, and in part, based on the vague assertion that the risks of deferral could exceed the benefits the closer a deadline gets. What exactly are the risks and benefits of deferral? and why would the risks of deferral exceed the benefits under high time pressure? The remaining hypotheses are equally underdeveloped, and in the case of H4, appears tautological (chronically indecisive people, by definition, have difficulty making decisions).

Thank you very much for your comment. We agree that previously described motivations were overly brief. To increase the rationales for our hypotheses we added further evidence and explanations. Furthermore, we now differentiate between our two forms of decision avoidance within the hypotheses.

To clarify H1 we added the following: ‘However, higher time pressure should reduce decision deferral, because the costs of deferral (e.g., risk that previous options change or that you lose options) could exceed possible benefits (e.g., gain of information or clarity) the closer a deadline gets. Under a condition of low time pressure, decision deferral could be reasonable to search for further information or alternatives or to increase deliberation about consequences and to capture the complexity (Milgram & Tenne, 2000). Besides of the expected gain of information, deferral has its added value by increasing certainty with decision and acceptance of unpleasant decisions (Eliaz & Schotter, 2010; Tykocinski & Ruffle, 2003). The relation between the length of time and the deferral and avoidance behavior has already been shown (Dhar & Nowlis, 1999; Tversky & Shafir, 1992; Tykocinski & Ruffle, 2003). Our study seeks to replicate these findings in further, complementing decision situations and tests for the first time whether the effects differ between decision deferral and option refusal. In there, larger effects are expected on decision deferral than on option refusal, because in the context of high time pressure decision deferral implies no gain, but option refusal could still be reasonable. (p10, line 421-433)

To complement motivation for H2 we added the following: ‘The completeness of the information mainly characterizes uncertainty in decision situations. Missing information should increase uncertainty and thereby selection difficulty because the options cannot be conclusively compared, and therefore it could remain open which option is superior. Decision avoidance could have the aim of reducing the lack of transparency by collecting further information (Corbin, 1980; Fischer, Greiff, & Funke, 2012; Greenleaf & Lehmann, 1995). This could be reasonable if the available time is sufficient (Milgram & Tenne, 2000). Avoiding decisions under uncertainty has been investigated within lottery tasks, like the Ellsberg Paradox (Ellsberg, 1961). As these tasks are far from reality, the association between uncertainty and decision avoidance should be investigated with more realistic decision tasks, as done in this study. Furthermore, this study firstly investigates whether the effects differ between decision deferral and option refusal. Due to the higher uncertainty, which can be concluded with more time, decision deferral seems more evident than option refusal under the condition of a high lack of information. Nevertheless, it is possible that if both options are associated with an as yet unclear risk, they are more likely to be rejected.’ (p. 10-11, line 436-465)

To clarify evidence for H3 we also added further evidence and explanations: ‘The nature of alternatives also influences decision-making. Decisions between two unattractive options (aversion-aversion conflict) are usually more difficult and more unpleasant than decisions between attractive options (appetence-appetence conflict; Lewin, 1936; Miller, 1944). Therefore, in decision situations of high stress, a decision between two unattractive options is often avoided (as studied in status quo bias and omission bias, Ritov & Baron, 1990; Samulson & Zeckhausen, 1988). Additionally, research about decisions in the context of critical incidents showed that redundant deliberation about negative consequences results in behavioral inaction and even worse outcomes (decision inertia, Power & Alison, 2018). On the other hand, it seems reasonable to continue searching for new alternatives if time pressure is low, and previous options are unsatisfying (Beach, 1993; Dhar, 1996). Therefore, in the context of consumer decisions, option refusal increases if alternatives are not attractive (Dhar, 1997b). In there, the avoidance seems like an adequate strategy and no dysfunctional avoidance. Therefore, this study investigates how the attractiveness of the choice-set influences the choice of decision deferral or option refusal in the context of daily decision situations. In there, larger effects are expected on option refusal than on decision deferral.’ (p. 11, line 468-481)

With regard to H4 we agree that the connection between chronic indecisiveness and decision avoidance is kind of obvious. However, it is necessary to empirically show the connection between self-reported indecisiveness and decision-making behavior in our vignettes. This has not always been consistent in previous studies (e.g., Patalano & Wengrovitz, 2007) To clarify our motivation we added the following:

‘Chronic indecisiveness is probably the most important personal factor identified in previous studies. Indecisive persons experience more difficulty and are less confident in their decision-making (Veinott, 2002). They have a higher threshold of feeling confident (Frost & Shows, 1993) and collect more information about the relevant alternatives (Ferrari & Dovidio, 2000). Therefore, indecisiveness is expected to be a predictor of deferring choice (Ferrari & Dovidio, 2000; Frost & Shows, 1993; Rassin & Muris, 2005). Patalano and Wengrovitz (2007) reported that indecisive participants did not show uniformly increased deferral, but rather that their deferral behavior is less strongly adapted to context conditions like increased risk. Therefore, this study wants to show if the indecisiveness of a person has an additional predictive effect compared to the situational factors in the prediction a decision avoidance behavior. Therefore, it is included as the main personal factor in the analysis models. It has never been studied before if chronic indecisiveness impacts decision deferral and options refusal as two options of decision avoidance differently. Since indecision is primarily associated with more time to decide, we expect it to be mainly associated with decision deferral and less with option refusal.‘ (p.12, line 492-504)

Beyond the main effects described in the current hypotheses, the theoretical development would greatly benefit from considering how the various personal and situational factors might interact with each other, instead of assuming a simple additive model.

We agree that beyond the main effects, the interaction between personal and situation factors would be interesting to further study. In this study, we firstly studied the effect of three main effects of the situation, one main effect of the person, and three further control variables on two different forms of decision avoidance. In there, considering all possible interactions would have gone beyond the scope. However, we already included this suggestion within our discussion section: ‘There are several recommendations for future studies. Besides personal and situational factors, the interaction of person and situation or different situational factors could be integrated by including mediators or moderators. For example, time pressure is assumed to be a moderator of other factors, so that some situational factors only influence decision avoidance in the condition of low time pressure but not in the condition of high time pressure.’ (p. 37-38, line 1345-1359)

To address your point at least to some extent, we added a table to the results section that showed all combinations of the manipulated situational factors to get an overview of possible interaction effects. In line with that, we added some information about interaction effects: ‘By looking at the different combinations (Table 2), only under the condition of an attractive choice-set and low time pressure, a high lack of information (code 221) implied more decision deferral than a low lack of information (code 211). This effect was consistent across the four situations.’ (p.24, line 824-827)

Furthermore, we have already addressed the interaction of attractivity*lack of information within the analyses. 

The conceptual and operational definitions of the main constructs are also problematic. For instance, the manuscript treats option refusal and decision deferral as distinct non-overlapping constructs. Yet choice deferral often implies option refusal. That is, people often defer choice in order to seek other alternatives. Also, the manuscript uses satisfaction with a decision as a proxy measure for decision quality, but decision quality is conceptually very different from satisfaction with the decision. One can be satisfied with a poor-quality decision or dissatisfied with a high-quality decision, especially when asked right after making the decision and before the consequences are known. Many of these problems can be addressed by clearly defining the constructs (e.g., what does decision quality really mean?), and ensuring that operational definitions (i.e., measurement) are consistent with conceptual definitions.

Thank you for this comment. As you stated, previous studies in consumer research often combined choice deferral and option refusal in a no-choice condition “not choose either of … and look for other…” (e.g., Dhar & Nowlis, 1999; Dhar, 1997, Dhar, 1996). However, we argue, in real life these options often don’t go hand in hand. You can refuse options without further searching so that you decide not to buy anything (option refusal without deferring). Or you can search for further options, but still keep in mind the ones you previously saw (deferring without option refusal). In our study, we separate these two forms of decision avoidance, so that a decision-maker has the possibility to choose, to defer choice (decision deferral), and to refuse choosing any option (option refusal). Decision deferral is defined as to postpone the decision to a later point in time, while option refusal is a decision against existing options. We agree that under some circumstances the two forms of decision avoidance overlap. If a decision-maker refuses current options and searches for further ones, the final decision is also deferred to a later point in time. The options are not completely distinct, but nevertheless, we believed that different factors of the situation and person contribute to the choice of either one or the other. Additionally, under different circumstances other forms of decision avoidance seem reasonable or dysfunctional avoidance. We were able to confirm this in our study.

To clarify the definition of our two forms of decision avoidance we included a paragraph within the introduction: ‘In real life, decisions are often deferred to a later point in time to collect more information (Corbin, 1980) or options are refused, because no alternative satisfies (Beach, 1993). Anderson defines choice deferral as a form of decision avoidance that is characterized by either postponing a decision or refusing to select an option (Anderson, 2003). Previous studies in consumer research often combined both options in a no-choice condition (not chose either of these and look for others; e.g., Dhar & Nowlis, 1999; Dhar, 1997, Dhar, 1996). Studies about conditions that affect different forms of decision avoidance are still lacking, even though Anderson (2003) had already pointed out this research gap 17 years ago. Therefore, we want to address this research gap by analyzing how situational and personal factors influence two different forms of the avoidance option (decision deferral and option refusal). Hence, in our study, we separate these two forms of decision avoidance, so that a decision-maker has the possibility to either choose, defer choice (decision deferral), or refuse to choose any option (option refusal). Decision deferral is defined as to postpone the decision to a later point in time, while option refusal is a decision against existing options. In line with this, two forms of decision avoidance were investigated in this study: (1) the option of deferring choice to a later point in time (decision deferral) and (2) the option to not decide at all by refusing both alternatives (option refusal).’ (p. 4-5, line 103-136)

To refer to the objection of overlap, we additionally included a paragraph within the limitation section: ‘Finally, decision deferral and option refusal are different forms of decision avoidance but not necessarily have to be distinct. However, our results showed that participants seem to experience the two forms as qualitatively different because there were clear intraindividual variations in the use of the form of avoidance option depending on situational variations in the vignettes.’ (p. 37, line 1328-1332)

To address the second point, we agree that decision quality is conceptually different from satisfaction with the decision. However, we believe that in real decisions, there are usually no best options so that normative measures for decision quality are inadequate and reasonable reasons for waiting are subjectively motivated. Therefore, we use the satisfaction as a subjective indicator for the quality of either the decision or the choose of an avoidance option. We now tried to better explain this with the following sentence: ‘The study aimed to analyze how participants experience their avoidance behavior to capture functional and dysfunctional avoidance out of a subjective perspective instead of providing predefined best options.’ (p.8, line 349-351) In order to better delimit our construct, we now added the term ‘subjective’: ‘Therefore, in this work, satisfaction with the decision-making behavior was used as subjective indicator for the quality of decision-making behavior.’ (p.9, line 385-387)

2. Has the statistical analysis been performed appropriately and rigorously?

There are concerns about aspects of the statistical analysis, reporting of the analysis, and interpretation of the findings.

First, not all interdependencies in the data were accounted for. While the analysis accounted for the fact that vignette-level factors were nested within respondents, it failed to account for the nesting within decision situations. The data were instead combined for all four decision situations.

In this study we used four different decision situations to better ensure generalizability. The main focus of the study was to analyze the effect of the within-subject manipulated factors. However, we agree that the effect of the four different situations are interesting as well. The multi-level model needed the aggregation of the four situations to ensure sufficient sample size. However, we agree that it is important to provide a better sense of how consistent the effects are. We supplemented a Table 2 (p. 25) that shows the eight variations across the four situations and added some information about the consistency of the manipulated situational effects within the manuscript. In there you see that the main effects of the manipulations were quite consistent across the situations: 

‘To get an overview of the four different decision situations and the interactions of the manipulated factors, see Table 2.’ (p.23, line 808-810)

‘This effect was consistent across the four situations (Table 2).’ (p. 24, line 819-820)

‘By looking at the different combinations (Table 2), only under the condition of an attractive choice-set and low time pressure, a high lack of information (code 221) implied more decision deferral than a low lack of information (code 211). This effect was consistent across the four situations.’ (p.24, line 824-827)

‘This effect was consistent across the four situations, but even more salient in the bigger decision situation 3 (internship) and 4 (apartment) than in decision situation 1 (seminar) and 2 (plans for evening) (Table 2).’ (p. 24, line 833-835)

Both the descriptive statistics (under the heading: Decision Avoidance Behavior) and the multi-level models fail to report the effect of decision situation (the between-subjects variable). How do the patterns of decision avoidance vary across the decision situations?

We agree that it is interesting how the patterns of decision avoidance vary across the decision situations. Therefore, we provided the new Table 2 (p. 25) to be transparent about how the pattern of decision avoidance varies across the decision situations (see also comment above). Additionally, we added some descriptive information about the frequency of different forms of decision avoidance across the decision situations under the heading “Decision Avoidance Behavoir”: 

‘Four different decisional situations adapted to a student’s everyday life had been studied: seminar at university, plans for the evening, internship, and student apartment. Decision deferral was most frequent when deciding on plans for the evening (M = 1.53, SD = 1.59) and least frequent when deciding on an apartment (M = 0.53, SD = 0.77). Option refusal was most frequent when deciding between two internships (M = 2.49, SD = 1.49) and least frequent when deciding on plans for evening (M = 0.62, SD = 0.91). To evaluate the effect of the within-subject manipulated factors, the four situations were aggregated to increase generalizability.’ (p.23, line 794-800)

The odds ratios are misinterpreted in several places in the manuscript. For instance, on page 11, referring to an OR of 48.09, it is stated that the probability of decision deferral was 48-times higher with low time pressure than with high time pressure. That is incorrect. The odds are 48-times higher not the probability. Similar statements are found on page 13. Also, OR < 1 are interpreted as very small effects instead of a reduction in the odds. OR < 1 means that the first group is less likely to experience the event than the second group. It doesn’t necessarily imply a very small effect. It would also be useful to report confidence intervals around the ORs.

Thank you for this comment. We agree that probability and odds were sometimes mixed up. Now, we have clarified our interpretation of the odds ratios. Additionally, we added confidence intervals around the ORs in Table 4 (p.28).

‘The strengths of evidence are presented as odds ratios (OR) and evaluated based on Jeffreys’ criteria (Jeffreys, 2003): OR=0-1.6, barely worth mentioning; OR=1.6-3.3, substantial; OR=3.3-5.0, strong; OR=5.0-6.6, very strong; OR>6.6, decisive.’ (p.21, line 747-748)

‘The odds ratio of decision deferral was 48-times higher with low time pressure than with high time pressure, F(2.027) =220.447, p< .001 (OR = 48.09).’ (p. 29, line 1078-1079)

‘The odds ratio of option refusal was 62-times higher within decisions between unattractive options-sets than between attractive option-sets, F(2.027) =227.432, p< .001 (OR = 61.93).’ (p. 29-30, line 1090-1097)

‘Nevertheless, the effect in this study was less decisive than other situational factors (OR = 0.59).’ (p.32-33, line 1179-1190)

Table 4: why not present the full table (all the conditions)?

We did as suggested (see page 30).

Table 5: the note under the table refers to odds ratios, but there are no odds ratios in the table.

You are right! Now, the reference has been removed. Because of the continuous dependent variable no odds ratios were provided here (see page. 31).

3. Have the authors made all data underlying the findings in their manuscript fully available?

Yes, the data was submitted with the manuscript. However, the analysis script is not available.

Thank you for this comment. We now have completed the data set by an additional sheet that describes the variables (S2). Furthermore, we added the syntax for the multi-level-models (S3).

4. Is the manuscript presented in an intelligible fashion and written in standard English?

The writing is at times ambiguous, and would benefit from a thorough edit/rewrite. In addition, in attempting to build a strawman, past research is sometimes misrepresented (e.g., when claiming that the decision-making literature considers choice deferral as dysfunctional, or that past research has not considered a combination of personal and situational factors).

Thank you for your comment. To ensure readability we have made up a thorough editing by a native English speaker. We are sorry, if the presentation of previous literature was misleading in some places due to condensations. That is because previous studies have explored decision avoidance out of the perspective of different and partly unlinked psychological disciplines. Therefore, e.g. in procrastination research choice deferral has been considered as dysfunctional by nature, while in other areas choice deferral has been considered less evaluative (e.g. consumer research). However, we think there is a general trend e.g., by looking at the literature on heuristics and biases to take a negative view on decision avoidance. To address your point, in our revised version, we have clarified considerably the theoretical background to understand the constructs and research we are referring to.

To clarify the different perspectives and research traditions we added a paragraph at the beginning of the introduction: ‘Decision avoidance is a phenomenon that has been studied in various forms and psychological disciplines (Anderson, 2003): choice deferral of consumer decisions in marketing research (e.g., Dhar, 1996), heuristic and biases of decision avoidance in cognitive psychology (e.g., Tversky & Shafir, 1992), decision inertia in critical incidents in naturalistic decision-making research (e.g., Power & Alison, 2018), procrastination research in educational or business psychology (e.g., Solomon & Rothblum, 1984), chronic indecisiveness in clinical psychology or personality psychology (e.g., Leykin & DeRubeis, 2010). Altogether, there is a bunch of studies in the context of decision avoidance, but they have explored the phenomenon out of the perspective of different and partly unlinked psychological disciplines. In line with Anderson (2003), the present study aimed to complement different research traditions to a more comprehensive view.’ (p. 3, line 45-54)

Furthermore, we agree that we are not the first researchers that combine personal and situational factors in studying decision avoidance. However, this perspective is rare and to our knowledge, there is no comparable study that considers the wealth of different influencing factors. Furthermore, no previous study analyzed the different effects of situational and personal factors on decision deferral and option refusal so far. We clarified the contribution of previous studies: ‘The combination of personal indecision and the risk of a decision has already been studied (Ferrari & Dovidio, 2000; Patalano & Wengrovitz, 2007). Nevertheless, studies integrating situational and personal factors in predicting the choice of an avoidance option are rare.’ (p. 5, line 141-144)

Comments of Reviewer #3: 

This paper presents a vignette-based decision making study focusing on decisional deferral (putting off a decision between options) and decisional avoidance (choosing neither option). As vignette studies go, the methods are generally rigorous, and some of the findings are interesting. The analyses seem to be appropriate and sound, but I am not an expert in the relevant statistical procedures.

We are pleased that our method seems sound and some results are interesting. 

I have three primary concerns that significantly undermine my enthusiasm for the paper. First, the paper is very difficult to read. My guess is that thorough editing by a native English speaker would help considerably, but I'm not sure whether simply clarifying the writing will fully resolve the problem. In any case, the writing made it difficult to fully evaluate both the novelty of the study and the precise methods used.

We are sorry about possible difficulties in reading. To ensure readability we have made up a thorough editing by a native English speaker.

Speaking of novelty, I couldn't quite figure out whether and to what extent the research questions and hypotheses were largely replicating past work or moving into new territory. The introduction cites numerous studies that would seem to address the authors' hypotheses, without making fully clear how the current study departs from or extends beyond that previous work.

Thank you for this comment. We markedly revised our hypotheses section to better explain the differences between previous literature and lack of research. Previous studies have never analyzed how situational and personal factors predict two forms of decision avoidance differently. Therefore, this is the first study that analyzes how decision deferral and option refusal are specifically experienced and influenced by different contextual conditions.

Therefore, we clarified: ‘Overall, this study aimed 1) to investigate the impact of personal and situational factors that influence decision deferral and option refusal as two forms of decision avoidance and 2) to evaluate, the experienced satisfaction of either choosing, deferring, or refusing depending on manipulated context conditions. ‘ (p.9, line 389-392) and ‘Our study aimed to differentiate previous findings for the two different forms of decision avoidance.’ (p.10, line 415-416)

For every hypothesis we added some information about our idea that fills a research gap:

H1: ‘Our study seeks to replicate these findings in further, complementing decision situations and tests for the first time whether the effects differ between decision deferral and option refusal. In there, we expect the effect to be greater on decision deferral than on option refusal, because in the context of high time pressure decision deferral implies no gain, but option refusal could still be reasonable. (p. 10, line 429-433) 

‘H1: Low time pressure implies more decision avoidance (especially decision deferral) than high time pressure.’ (p. 10, line 434-435) 

H2: ‘Furthermore, this study firstly investigates whether the effects differ between decision deferral and option refusal. Due to the higher uncertainty, which can be concluded with more time, decision deferral seems more evident than option refusal under the condition of a high lack of information. Nevertheless, it is possible that if both options are associated with an as yet unclear risk, they are more likely to be rejected.’ (p. 10, line 434-435) 

‘H2: Lack of information implies more decision avoidance (especially decision deferral) than full information.’ (p. 11, line 466-467)

‘Therefore, in the context of consumer decisions, option refusal increases if alternatives are not attractive (25). In there, the avoidance seems like an adequate strategy and no dysfunctional avoidance. Therefore, this study investigates how the attractiveness of the choice-set influences the choice of decision deferral or option refusal in the context of daily decision situations. In there, larger effects are expected on option refusal than on decision deferral.’ (p. 11, line 476-481)

‘H3: Decisions within an unattractive choice-set imply more decision avoidance (especially option refusal) than decisions within an attractive option-set.’ (p. 11-12, line 482-490)

H4: ‘Patalano and Wengrovitz reported that indecisive participants did not show uniformly increased deferral, but rather that their deferral behavior is less strongly adapted to context conditions like increased risk. Therefore, this study wants to show if the indecisiveness of a person has an additional predictive effect compared to the situational factors in the prediction a decision avoidance behavior. Therefore, it is included as the main personal factor in the analysis models. It has never been studied before, if chronic indecisiveness impacts decision deferral and options refusal as two options of decision avoidance differently. Since indecision is essentially associated with more time to decide, we expect it to be mainly associated with decision deferral and less with option refusal.’ (p. 12, line 496-504) 

‘H4: The higher the indecisiveness of a person, the more decision avoidance is shown (especially decision deferral).’ (p. 12, line 505-506)

Finally, and most problematically, I am simply underwhelmed by studies of decision-making "behavior" that rely on hypothetical decisions based on simple vignettes. I certainly understand that this approach allows each participant to consider scenarios that vary on a number of dimensions; however, this strength isn't very useful if people's decisions fail to reflect decisions they would make in real life. Many decades of social psychological research has confirmed that people are pretty terrible at predicting how they would act in a hypothetical scenario.

Thank you for your comment. We are aware of the possibility of a mismatch between decision-making behavior in hypothetical scenarios and real-life behavior. However, to ensure practicability and causality, experimental vignette methodology is a useful way to address aspects of both internal versus external validity (Aguinis & Bradley, 2014). In line with that there is also some evidence of no difference between choices in real and hypothetical consequence conditions (e.g., (Wiseman & Levin, 1996). We believe that the mismatch is especially a problem with moral decisions (social desirability) or high-risk decisions. Therefore, we highly value research in the field of naturalistic decision research that studies real world behavior of experts in complex and high stress situations. In our study of everyday decisions in a student’s life, it is much easier to put yourself in the situation as you are familiar with it. We have attached particular importance to selecting decision situations that are of relevance and that are formulated in a realistic way and fit to our sample of student participants.

To ensure external validity, we have assessed how realistically participants experienced the vignettes. The results showed that participants rated the decision situations to be rather realistic and that they are quite familiar with similar situations. Besides, decision situations were clearly not considered trivial as they were rated as important and moderately difficult. We now added this data within the manuscript. Furthermore, we included a whole paragraph about realistic decision-making scenarios in the discussion.

In the method section: ‘To ensure external validity the following was captured: Participants were asked as to how realistically they experienced the decision situation and if they had their own experiences with a similar decision situation in the past (one-item scales of 0-100).’ (p.20, line 714-717)

In the results section: ‘The participants rated the decision situations to be rather realistic (M = 70.13, SD = 22.28; scale 0-100). Some of the participants had already had their own experiences with a similar situation in the past (M = 45.82, SD = 30.88; scale 0-100). On average, decision situations were assessed as rather important (M = 70.60, SD = 22.76; scale 0-100) and moderately difficult (M = 42.00, SD = 21.18; scale 0-100).’ (p.22, line 764-769)

In the discussion: ‘This study takes up some concerns that naturalistic decision-making research has about classic decision-making research (e.g., Lipshitz, Klein, Orasanu et al., 2001): The vignettes reflect familiar decisional situations that are formulated in a realistic way and fit our sample of student participants. Additionally, the study focuses on process-orientation in contrast to input-output-orientation by analyzing when people chose an avoidance option and how satisfied they are with it instead of providing predefined best choices. Nevertheless, this study evaluated hypothetical decision situations and no real-life behavior to systematically vary different dimensions to evaluate the importance of the prediction of an avoidance option. With this approach, we are in line with Todd and Gigerenzer (2001) that argue no general contradiction between formal modeling and descriptively valid and task-specific real-world decisions. Therefore, we consider important real-life context conditions like time pressure and incompleteness of the information but formalize them in standardized vignettes to combine different advantages.’ (p.36, line 1298-1309)

Nevertheless, we agree that hypothetical scenarios provide some risks. Therefore, we already had stated within the limitation section: ‘The present study was conducted with self-developed vignettes so that results could deviate from real decision-making behavior (Groß & Börensen, 2009). It can be assumed that decision avoidance is of greater importance in real-life decision situations and the possibility of socially desirable responses has to be taken into account because decision deferral is considered a weakness. Nevertheless, vignette studies provide high internal validity and allow to hierarchically assess personal and situational factors (Atzmüller & Steiner, 2010).’ (p.36-37, line 1310-1320) 

However, we are still convinced of the advantages as we state ‘Finally, decision situations were described in a realistic way and stimulus material and student sample constituted a good fit. This was also supported by our participants that assessed the decision situations to be rather realistic. Overall, this strengthens external validity.’ (p.37, line 1341-1344)

As a minor comment, I was surprised that the authors collapsed across their 4 scenario domains without testing whether effects differed across them. The evidence would be somewhat more compelling if the authors instead treated the 4 scenarios as four substudies and then conducted mini-meta-analyses of the effects across those substudies to provide a better sense of how consistent the effects are.

Thank you for this comment. We agree that it is interesting how the effects probably differ across the 4 scenarios. As our main interest was not on the different scenarios, but on the other factors of the situation and person, we aggregated the scenarios for generalization. This was also to provide sufficient sample size for the multi-level analyses we conducted. However, we agree that it is important to provide a better sense of how consistent the effects are. Therefore, we now provide data about how decision deferral and option refusal vary across the 4 situations: 

‘Four different decisional situations adapted to a student’s everyday life had been studied: seminar at university, plans for the evening, internship, and student apartment. Decision deferral was most frequent when deciding on plans for the evening (M = 1.53, SD = 1.59) and least frequent when deciding on an apartment (M = 0.53, SD = 0.77). Option refusal was most frequent when deciding between two internships (M = 2.49, SD = 1.49) and least frequent when deciding on plans for evening (M = 0.62, SD = 0.91). To evaluate the effect of the within-subject manipulated factors, the four situations were aggregated to increase generalizability.’ (p.23, line 94-100)

Furthermore, we supplemented a Table 2 (p. 25) that shows the eight variations across the four situations and added some information about the consistency of the manipulated situational effects within the manuscript. 

‘To get an overview of the four different decision situations and the interactions of the manipulated factors, see Table 2.’ (p.23, line 808-810)

‘This effect was consistent across the four situations (Table 2).’ (p. 24, line 819-820)

‘By looking at the different combinations (Table 2), only under the condition of an attractive choice-set and low time pressure, a high lack of information (code 221) implied more decision deferral than a low lack of information (code 211). This effect was consistent across the four situations.’ (p.24, line 824-827)

‘This effect was consistent across the four situations, but even more salient in the bigger decision situation 3 (internship) and 4 (apartment) than in decision situation 1 (seminar) and 2 (plans for evening) (Table 2).’ (p. 24, line 833-835)

In short, I fear that the methodological shortcomings may be difficult to overcome in a revision. However, due to problems with clarity, it is possible that the authors could convince me of the novelty of their endeavor.

Thank you very much to providing us the opportunity to convince you about the novelty of our endeavor. We hope that our markedly revisions revealed more clarity.

---

## [Decision Letter · Decision Letter 1]

2 Apr 2020

PONE-D-19-17779R1

A vignette study of option refusal and decision deferral as two forms of decision avoidance: Situational and personal predictors

PLOS ONE

Dear Dr. Funke,

Thank you for submitting your manuscript to PLOS ONE. After careful consideration, we feel that it has merit but does not fully meet PLOS ONE’s publication criteria as it currently stands. Therefore, we invite you to submit a revised version of the manuscript that addresses the points raised during the review process.

We would appreciate receiving your revised manuscript by May 17 2020 11:59PM. To enhance the reproducibility of your results, we recommend that if applicable you deposit your laboratory protocols in protocols.io, where a protocol can be assigned its own identifier (DOI) such that it can be cited independently in the future. For instructions see: http://journals.plos.org/plosone/s/submission-guidelines#loc-laboratory-protocols

We look forward to receiving your revised manuscript.

Kind regards,

Valerio Capraro

Academic Editor

PLOS ONE

Additional Editor Comments (if provided):

I have now collected two reviews from two of the same three experts who reviewed the first version of this paper (the third reviewer declined to review the revised version because of lack of time). While one review is positive, the other one reports a conceptual flaw that, in the reviewer's view, cannot be addressed in a revision. Since the other review is positive, as it was the third initial review, I would like to give you the opportunity to revise the manuscript and thus respond to the negative reviewer and try to address this conceptual flaw, perhaps through a careful rewriting of the manuscript. Needless to say that this revision should be done, indeed, in a very careful way.

Looking forward for the revision.

And I take this occasion to wish you and your loved ones to be and remain healthy and safe during these difficult times.

Reviewers' comments:

Reviewer's Responses to Questions

**Comments to the Author**

1. If the authors have adequately addressed your comments raised in a previous round of review and you feel that this manuscript is now acceptable for publication, you may indicate that here to bypass the “Comments to the Author” section, enter your conflict of interest statement in the “Confidential to Editor” section, and submit your "Accept" recommendation.

Reviewer #2: (No Response)

Reviewer #3: (No Response)

2. Is the manuscript technically sound, and do the data support the conclusions?

Reviewer #2: No

Reviewer #3: Yes

3. Has the statistical analysis been performed appropriately and rigorously? 

Reviewer #2: No

Reviewer #3: Yes

4. Have the authors made all data underlying the findings in their manuscript fully available?

Reviewer #2: Yes

Reviewer #3: No

5. Is the manuscript presented in an intelligible fashion and written in standard English?

Reviewer #2: Yes

Reviewer #3: No

6. Review Comments to the Author

Reviewer #2: The authors have done a commendable job revising the manuscript, and have addressed several limitations raised in the first round of reviews. However, significant concerns remain, and unfortunately, I do not believe they can be easily addressed with a new revision.

The main contribution of this research is in examining the conditions that affect different forms of decision avoidance. This can only be achieved if the research indeed investigates two distinct forms of decision avoidance. However, choice deferral and option refusal overlap to a great extent. When people defer their choice, it is often because they are not satisfied with the existing options. In their response to this criticism, the authors acknowledged the overlap, but argued that people can refuse options without deferring the decision (e.g., deciding not to buy anything). This conceptualization of option refusal, however, is hardly a form of decision avoidance. On the contrary, it is an explicit decision to reject the existing options. In other words, deciding not to buy anything is different from not deciding which option to buy or whether to buy or not.

Throughout the paper, option refusal is conflated with refusal to choose/select, but as mentioned above, when people reject the alternatives they are presented with, they are making an explicit decision. They are not avoiding to decide.

If we adopt the authors’ conceptualization of option refusal as independent from deferral, then we have to accept that option refusal is not a form of decision avoidance, and therefore, this study does not in fact examine the factors that affect different forms of decision avoidance as purported.

Other comments:

Presenting descriptive statistics for the four situations is a good start, but the multilevel analyses (and thus the hypothesis tests) still do not take all the dependencies in the data into account. The argument that aggregating across the situation was necessary to ensure sufficient sample size only indicates that the study was inadequately planned. Regardless, there are other ways to conduct the analysis. For example, mixed effect models are mathematically equivalent to multilevel models, but are more flexible and would allow you to specify random intercepts and random slopes for situations.

There are several statements about differences and similarities between the four situations (e.g., page 23, lines 523-526; page 24, lines 543-544; page 24, line 551; page 24, lines 557-559) that are not based on any statistical tests. They seem to be only based on eye balling the data summarized in table 2.

The writing, although improved, would still benefit from a thorough edit/rewrite.

Best of luck with your project!

Reviewer #3: This paper presents a vignette-based decision making study focusing on decisional deferral (putting off a decision between options) and decisional avoidance (choosing neither option). I reviewed the previous version of this manuscript. At that time, my primary concerns were with clarity, novelty, and methodology. I’m pleased to note that the authors have improved the paper on all fronts. The fluency of the writing is much better (although still a bit rough in places), and the introduction makes clearer how the present study goes beyond previous evidence. Although we’re inevitably stuck with the hypothetical scenarios, which were the source of my main methodological critique, I understand the benefits of such an approach. The methodology otherwise remains quite strong, and the findings are interesting.

My only remaining suggestion is to streamline the introduction a bit. It’s quite long, and it feels redundant in places. At the very least, integrating the hypotheses into the first sections on situational and personal factors, rather than essentially repeating those sections to introduce the hypotheses, would be a considerable improvement.

As a final note, I did not see evidence in the paper that the data are publicly available, as required by this journal. Apologies if I missed it.

7. PLOS authors have the option to publish the peer review history of their article (what does this mean?). If published, this will include your full peer review and any attached files.

Reviewer #2: No

Reviewer #3: No

---

## [Author Response · Author response to Decision Letter 1]

22 May 2020

Rebuttal Letter

Comments of the Editor

I have now collected two reviews from two of the same three experts who reviewed the first version of this paper (the third reviewer declined to review the revised version because of lack of time). While one review is positive, the other one reports a conceptual flaw that, in the reviewer's view, cannot be addressed in a revision. Since the other review is positive, as it was the third initial review, I would like to give you the opportunity to revise the manuscript and thus respond to the negative reviewer and try to address this conceptual flaw, perhaps through a careful rewriting of the manuscript. Needless to say that this revision should be done, indeed, in a very careful way.

Looking forward for the revision.

Answer: Thank you very much for giving us the opportunity to address the conceptual flaw in the view of reviewer 2 in a rebuttal. We hope that we have addressed all critical issues so that this manuscript can be accepted.

Reviewer #2: 

R2.1 The authors have done a commendable job revising the manuscript, and have addressed several limitations raised in the first round of reviews. However, significant concerns remain, and unfortunately, I do not believe they can be easily addressed with a new revision.

Answer: Thank you very much for appreciating our markedly revisions. We hope that we could provide further improvements and clarifications with regard to the remaining concerns.

R2.2 The main contribution of this research is in examining the conditions that affect different forms of decision avoidance. This can only be achieved if the research indeed investigates two distinct forms of decision avoidance. However, choice deferral and option refusal overlap to a great extent. When people defer their choice, it is often because they are not satisfied with the existing options. In their response to this criticism, the authors acknowledged the overlap, but argued that people can refuse options without deferring the decision (e.g., deciding not to buy anything). This conceptualization of option refusal, however, is hardly a form of decision avoidance. On the contrary, it is an explicit decision to reject the existing options. In other words, deciding not to buy anything is different from not deciding which option to buy or whether to buy or not.

Throughout the paper, option refusal is conflated with refusal to choose/select, but as mentioned above, when people reject the alternatives they are presented with, they are making an explicit decision. They are not avoiding to decide.

If we adopt the authors’ conceptualization of option refusal as independent from deferral, then we have to accept that option refusal is not a form of decision avoidance, and therefore, this study does not in fact examine the factors that affect different forms of decision avoidance as purported.

Answer: Thank you very much for this comment. We hope to provide more clarity of our conceptualization by classifying it within the literature. Anderson (2003) defines decision avoidance as follows: “Decision avoidance manifests itself as a tendency to avoid making a choice by postponing it or by seeking an easy way out that involves no action or no change“. In there, he distinguishes between postponing (that we label decision deferral) and no selection/no change (this could be to refuse options, or to decide for status quo or to omit an action). Therefore, it is possible to distinguish between decision deferral and option refusal as two different forms of decision avoidance. We operationalize the two forms as follows: (1) ‘I will decide at a later time’ (decision deferral) that leaves the decision-making process open and (2) ‘I choose none of the alternatives’ (option refusal) that is at least a decision against the existing options. 

Nevertheless, we understand that the term ‚decision avoidance’ can easily be misunderstood to mean only a passive avoidance and not an active decision to avoid a decision/selection by refusing current options. One reason for this is that it sounds paradoxical from the term that avoiding a decision can also be an active decision. Another reason is that this understanding is also reflected in some research traditions: E.g., in Janis and Mann's (1977) conflict theory of decision making, procrastination and buck-passing are conceptualized as defensive avoidance. ‘Defensive avoidance’ implies a negative valuation and is therefore per se a passive and dysfunctional form of decision avoidance. However, other research traditions cover a broader and less evaluative understanding of decision avoidance (e.g., Anderson). The fact that someone has actively chosen a refusal of options does not make it less of a "decision avoidance", but only an active and intended avoidance. Therefore, Anderson (2006) suggested to distinguish different forms of decision avoidance by the factor’s awareness of options, avoidant intention vs. seeking intention and active vs. passive (Anderson, 2006). In there, to defer a choice and to avoid a change could both be an active and deliberative decision or more a passive omission or difficulty to come to a decision. 

Overall, in our study, we refer to a broader and more neutral definition of decision avoidance that is also covered in the more recent research traditions (e.g., Anderson). Therefore, we integrate active and passive forms of decision avoidance. To further improve on a clear conceptualization, we added a section within the introduction:

‘The different forms of decision avoidance need not to reflect an inactive and passive process, but could be distinguished by avoidant intention vs. seeking intention in an active or passive way (Anderson, 2006): In the case of an avoidant intention, it could be a deliberate, but passive choice to do nothing (omission), but also a deliberate choice to take an action that results in no change (status quo). In the case of a seeking intention, there could be a difficulty in committing to a choice that results in a delay (procrastination), but it could also be an active and deliberate process to defer a decision (choice deferral).’ (p.3, line 54-60)

R2.3 Other comments:

Presenting descriptive statistics for the four situations is a good start, but the multilevel analyses (and thus the hypothesis tests) still do not take all the dependencies in the data into account. The argument that aggregating across the situation was necessary to ensure sufficient sample size only indicates that the study was inadequately planned. Regardless, there are other ways to conduct the analysis. For example, mixed effect models are mathematically equivalent to multilevel models, but are more flexible and would allow you to specify random intercepts and random slopes for situations.

Answer: As the focus of our study was on the systematically manipulated factors of the situation as well as the personal factors, we planned our study design and analyses accordingly. To justify the generalizability by our aggregation, we have already provided the descriptive data of the four situations. Nevertheless, we agree that an evaluation of the specific effects of the decision situations could be interesting for future studies, where this focus could be explicitly planned. 

To clarify our approach, we integrated the following within the method section: ‘The effect of the decision situation was no primary focus in the study and therefore not integrated as a separate factor in the model. Nevertheless, descriptive results of the four decision situations with regard to variations in decision deferral and option refusal depending on the manipulated factors of the situation were provided.’ (p.20, line 980-984)

Additionally, we integrated a sentence in the discussion to encourage future studies to put a focus on the effect of the decision situation: ‘To evaluate the specific effects of the decision situations further studies could integrate them as a separate factor within the multilevel models.’ (p.38, line 1378-1380)

R2.4 There are several statements about differences and similarities between the four situations (e.g., page 23, lines 523-526; page 24, lines 543-544; page 24, line 551; page 24, lines 557-559) that are not based on any statistical tests. They seem to be only based on eye balling the data summarized in table 2.

Answer: Thank you for this comment. We agree that these statements are just descriptive as we provide no statistical test results. To clarify this and to avoid giving the wrong impression, we revised the sentences: 

• ‘By descriptively evaluating the four decision situations separately (Table 2), this effect appears to be consistent across the four situations.’ (p.23, line 1045-1047)

• ‘This effect seems to be consistent across the four situations (Table 2).’ (p.23, line 1054-1055)

• ‘This effect seems to be approximately consistent across the four situations, but even more salient in the bigger decision situation 3 (internship) and 4 (apartment) than in decision situation 1 (seminar) and 2 (plans for evening) (Table 2).’ (p.23, line 1061-1063)

R2.5 The writing, although improved, would still benefit from a thorough edit/rewrite.

Answer: Thank you for appreciating improvement, we already revised the manuscript with the help of a Native Speaker. Additionally, we further reedited some issues. 

R2.6 Best of luck with your project!

Answer: Thank you very much!

 

Reviewer #3: 

R3.1 This paper presents a vignette-based decision making study focusing on decisional deferral (putting off a decision between options) and decisional avoidance (choosing neither option). I reviewed the previous version of this manuscript. At that time, my primary concerns were with clarity, novelty, and methodology. I’m pleased to note that the authors have improved the paper on all fronts. The fluency of the writing is much better (although still a bit rough in places), and the introduction makes clearer how the present study goes beyond previous evidence. Although we’re inevitably stuck with the hypothetical scenarios, which were the source of my main methodological critique, I understand the benefits of such an approach. The methodology otherwise remains quite strong, and the findings are interesting.

Answer: We are very pleased that our revisions with clarity, novelty, and methodology are convincing. Thank you for praising our methodology as strong and findings as interesting. 

R3.2 My only remaining suggestion is to streamline the introduction a bit. It’s quite long, and it feels redundant in places. 

Answer: Thank you for this comment. We shortened the introduction by abbreviating the redundancies and merging the sections about the aims of the study, research questions and hypotheses as well as the different sections on situational and personal factors. 

R3.3 At the very least, integrating the hypotheses into the first sections on situational and personal factors, rather than essentially repeating those sections to introduce the hypotheses, would be a considerable improvement.

Answer: Thank you for this idea. We restructured the introduction so that hypotheses are now integrated into the first sections on situational and personal factors. 

R3.4 As a final note, I did not see evidence in the paper that the data are publicly available, as required by this journal. Apologies if I missed it.

Answer: We are sorry, if data availability was unclear. Of course, we considered data availability and had already included the files as a supplement. To clarify this, we have now integrated a reference in the method section: 

‘To ensure data availability, the data set with all relevant variables in this study (S1), an overview of the variable labels, and the syntax of the multi-level models (S3) are provided in the supplement. For ethical reasons, sociodemographic factors are reported anonymous in a separated file, but this does not affect the reproducibility of the results.’ (p.20-21, line 986-990).

---

## [Decision Letter · Decision Letter 2]

26 Aug 2020

PONE-D-19-17779R2

A vignette study of option refusal and decision deferral as two forms of decision avoidance: Situational and personal predictors

PLOS ONE

Dear Dr. Funke,

Thank you for submitting your manuscript to PLOS ONE. After careful consideration, we feel that it has merit but does not fully meet PLOS ONE’s publication criteria as it currently stands. Therefore, we invite you to submit a revised version of the manuscript that addresses the points raised during the review process.

Please find the reviewers' comments, as well as those of mine, below.

We look forward to receiving your revised manuscript.

Kind regards,

Valerio Capraro

Academic Editor

PLOS ONE

Additional Editor Comments (if provided):

I am very sorry for the delay in making a decision regarding this manuscript. As you will see from the reviews below, the two reviewers who reviewed the previous version of this manuscript gave conflicting recommendations: one suggested to accept the manuscript, the other one to reject it. To solve this conflict, I have then invited a third reviewer. It was not easy to find a third reviewer during the covid pandemic, especially because reviewers are typically reluctant to review a paper that has already gone through a revision round. This is why it took so long. However, the good news is that the third review is positive. While s/he acknowledges the limitations found by the negative reviewer, s/he thinks that the paper can still be published after a minor revision. Therefore, I would like to invite you to revise your manuscript again. Please address all the comments at the best of your abilities. It is unlikely that I will send this manuscript out for review again, therefore, please do your best to put it in final form.

I am looking forward for the revision.

Reviewers' comments:

Reviewer's Responses to Questions

**Comments to the Author**

1. If the authors have adequately addressed your comments raised in a previous round of review and you feel that this manuscript is now acceptable for publication, you may indicate that here to bypass the “Comments to the Author” section, enter your conflict of interest statement in the “Confidential to Editor” section, and submit your "Accept" recommendation.

Reviewer #2: (No Response)

Reviewer #3: All comments have been addressed

Reviewer #4: (No Response)

2. Is the manuscript technically sound, and do the data support the conclusions?

Reviewer #2: Partly

Reviewer #3: (No Response)

Reviewer #4: Yes

3. Has the statistical analysis been performed appropriately and rigorously? 

Reviewer #2: No

Reviewer #3: (No Response)

Reviewer #4: Yes

4. Have the authors made all data underlying the findings in their manuscript fully available?

Reviewer #2: Yes

Reviewer #3: (No Response)

Reviewer #4: Yes

5. Is the manuscript presented in an intelligible fashion and written in standard English?

Reviewer #2: Yes

Reviewer #3: (No Response)

Reviewer #4: Yes

6. Review Comments to the Author

Reviewer #2: Once again, the authors have made serious efforts to revise the manuscript, which resulted in a more readable paper. I raised two major issues in the last round and, unfortunately, neither has been addressed adequately.

The first issue was conceptual and referred to the adequacy of conceptualizing option rejection as a form of decision avoidance. In their rebuttal, the authors argued that their (broader) definition of decision avoidance can accommodate option refusal. However, making arbitrary changes to conceptual boundaries is not a convincing argument. I will not belabour the point here, but option refusal is still conflated with refusal to choose/select.

The second issue concerned the analytical strategy used in the paper. The authors seem to have missed my point completely. In their reply, they argue that they aggregated the data because they are not interested in examining the differences between the scenarios. My point wasn't about the scope of the theory or what questions should be of interest. It was pointing a methodological/statistical flaw. Analyzing the aggregated data the way they did/still do violates the assumption of independence of error terms underlying their statistical model.

In sum, the research is not without merit, but the remaining conceptual and analytical problems make it difficult to draw any valid conclusions from the findings.

Reviewer #3: (No Response)

Reviewer #4: This is a revision of a study to investigate decision avoidance and to develop a more comprehensive view of types and facets of decision avoidance and their predictors. The initial set of revisions resulted in a conflict between the two reviewers. While the study has weaknesses, I do not believe that it has fatal flaws outlined by one of the reviewers. For instance, I am unsure I can agree with the reviewer that explicit refusal to decide (e.g., refusal to buy anything) is not a form of decision avoidance. While the reviewer may be correct that it is not an avoidance of decisions overall, it is a way to avoid the particular decision the person is facing. Another way of thinking about it is that if the initial decision was to choose between two options, a decision-maker effectively creates a 3rd option by refusing both of the original options, thereby avoiding the initial decision. As such, I believe that this study represents a contribution. The questions under investigation are interesting, the sample size is considerable, and analyses are comprehensive. However, I have several relatively minor concerns, as outlined below.

1. The authors state: “studies within clinical or educational 73 psychology do often classify choice deferral as a subdomain of procrastination”. That’s not true. In psychopathology indecisiveness is about either anxiety and the desire to avoid anxiety, or lack of trust in oneself to make a good option, or inability to make a choice which results in “giving up”. Neither of these are about procrastination.

2. In the introduction (lines 204 and on) the authors seem to conflate dual process theory and decision styles. These are not the same. Decision styles are patterns of decisions a person tends to rely on more often. Dual process theory posits that in different circumstance, people tend to engage in more intuitive or in more deliberative processes.

3. The authors posit that, following NDM approaches, normative models do not apply, and that decision satisfaction is the best measure of decision-making. It is unclear whether NDM applies to the types of the vignettes used by the authors (the situations are not exactly high-stakes or multifaceted, for instance). Also, models such as expected utility theory are usually considered normative and should apply in these cases. Further, decision satisfaction is often conflated with outcomes or decision-maker’s internal states (e.g., anxiety, depression). It seems that the authors are overstating the field when they say that degree of satisfaction is an established measure of decision quality. It is one measure, but it does not necessarily measure quality nor is it established.

4. Because odds ratios are expressed in the units of the predictor, does the Jeffreys’ criteria only apply when predictors are dichotomous?

5. The “time too fast (less than five times the median)” is not clear. Can the authors rephrase?

6. There is some awkwardness in phrasing that is apparent at many points, but it is not distracting.

7. The manuscript is at times unnecessarily detailed and repetitive; a considerable reduction would improve readability.

7. PLOS authors have the option to publish the peer review history of their article (what does this mean?). If published, this will include your full peer review and any attached files.

Reviewer #2: No

Reviewer #3: No

Reviewer #4: No

---

## [Author Response · Author response to Decision Letter 2]

9 Oct 2020

Comments of the Editor

I am very sorry for the delay in making a decision regarding this manuscript. As you will see from the reviews below, the two reviewers who reviewed the previous version of this manuscript gave conflicting recommendations: one suggested to accept the manuscript, the other one to reject it. To solve this conflict, I have then invited a third reviewer. It was not easy to find a third reviewer during the covid pandemic, especially because reviewers are typically reluctant to review a paper that has already gone through a revision round. This is why it took so long. However, the good news is that the third review is positive. While s/he acknowledges the limitations found by the negative reviewer, s/he thinks that the paper can still be published after a minor revision. Therefore, I would like to invite you to revise your manuscript again. Please address all the comments at the best of your abilities. It is unlikely that I will send this manuscript out for review again, therefore, please do your best to put it in final form.

I am looking forward for the revision.

-> Thank you very much for your effort and for giving us the opportunity to address the further comments in another rebuttal. We hope that we have addressed all critical issues so that this manuscript can be finally accepted.

 

Reviewer #2: 

Once again, the authors have made serious efforts to revise the manuscript, which resulted in a more readable paper. I raised two major issues in the last round and, unfortunately, neither has been addressed adequately.

-> Thank you very much for appreciating our markedly revisions. We hope that we could provide further improvements and clarifications with regard to the remaining concerns.

The first issue was conceptual and referred to the adequacy of conceptualizing option rejection as a form of decision avoidance. In their rebuttal, the authors argued that their (broader) definition of decision avoidance can accommodate option refusal. However, making arbitrary changes to conceptual boundaries is not a convincing argument. I will not belabour the point here, but option refusal is still conflated with refusal to choose/select.

We regret that we could not convince you with our conceptualization. In the previous revision we have explained our definition of decision avoidance and the difference between option refusal in detail and also adapted the manuscript accordingly. Therefore, we do not repeat all again, because we probably have different opinions here.

The second issue concerned the analytical strategy used in the paper. The authors seem to have missed my point completely. In their reply, they argue that they aggregated the data because they are not interested in examining the differences between the scenarios. My point wasn't about the scope of the theory or what questions should be of interest. It was pointing a methodological/statistical flaw. Analyzing the aggregated data the way they did/still do violates the assumption of independence of error terms underlying their statistical model.

We agree that we have argued here in terms of content and not in terms of statistics. We have understood the statistical objection, but do not consider it serious at this point. We justify this by the separate descriptive evaluations, where the effects appeared to be consistent across the four situations 

In sum, the research is not without merit, but the remaining conceptual and analytical problems make it difficult to draw any valid conclusions from the findings.

-> We are pleased that the reviewer sees the merit of our research. However, contrary to his/her opinion, we are also convinced that relevant conclusions can be drawn from the study.

 

Reviewer #4:

This is a revision of a study to investigate decision avoidance and to develop a more comprehensive view of types and facets of decision avoidance and their predictors. The initial set of revisions resulted in a conflict between the two reviewers. While the study has weaknesses, I do not believe that it has fatal flaws outlined by one of the reviewers. For instance, I am unsure I can agree with the reviewer that explicit refusal to decide (e.g., refusal to buy anything) is not a form of decision avoidance. While the reviewer may be correct that it is not an avoidance of decisions overall, it is a way to avoid the particular decision the person is facing. Another way of thinking about it is that if the initial decision was to choose between two options, a decision-maker effectively creates a 3rd option by refusing both of the original options, thereby avoiding the initial decision. As such, I believe that this study represents a contribution. The questions under investigation are interesting, the sample size is considerable, and analyses are comprehensive. However, I have several relatively minor concerns, as outlined below.

-> Thank you very much for your detailed discussion of the study. We are pleased that the reviewer sees the contribution of the study and finds the research interesting and the analysis comprehensive. We have put effort to adapt the minor concerns raised at the best of our abilities. 

1. The authors state: “studies within clinical or educational 73 psychology do often classify choice deferral as a subdomain of procrastination”. That’s not true. In psychopathology indecisiveness is about either anxiety and the desire to avoid anxiety, or lack of trust in oneself to make a good option, or inability to make a choice which results in “giving up”. Neither of these are about procrastination.

-> Thank you very much for this clarification. We agree that we have generalized somewhat imprecisely. In fact, educational psychology often refers to procrastination when it comes to postponing papers and exams, while clinical psychology is more concerned with increased anxiety, depressed lack of motivation or lack of self-confidence. We have adjusted this accordingly, see p. 4:

“In clinical psychology, indecisiveness is often considered as a form of anxiety, avoidance or lack of motivation or self-confidence. Studies within educational psychology do often classify choice deferral as a subdomain of procrastination.” (p. 4, line 72-74)

2. In the introduction (lines 204 and on) the authors seem to conflate dual process theory and decision styles. These are not the same. Decision styles are patterns of decisions a person tends to rely on more often. Dual process theory posits that in different circumstance, people tend to engage in more intuitive or in more deliberative processes.

-> Thank you very much for this comment. In our study we refer to decision styles. In order not to confuse, we have removed the reference to dual-process theory, see p. 9:

“Maybe the most classical distinction is the one between rational and intuitive decision-making.” (p. 9, line 207).

3. The authors posit that, following NDM approaches, normative models do not apply, and that decision satisfaction is the best measure of decision-making. It is unclear whether NDM applies to the types of the vignettes used by the authors (the situations are not exactly high-stakes or multifaceted, for instance). Also, models such as expected utility theory are usually considered normative and should apply in these cases. Further, decision satisfaction is often conflated with outcomes or decision-maker’s internal states (e.g., anxiety, depression). It seems that the authors are overstating the field when they say that degree of satisfaction is an established measure of decision quality. It is one measure, but it does not necessarily measure quality nor is it established.

-> Thank you very much for this comment. We agree that we did not use classical NDM vignettes, because we missed high-stakes for example. However, we conceptualized descriptively valid and task-specific real-world decisions that consider important naturalistic context conditions like time pressure and incompleteness of the information. Additionally, the study focuses on process-orientation in contrast to input-output-orientation by analyzing when people chose an avoidance option and how satisfied they are with it instead of providing predefined best choices. That is the main reason, why we focused on decision satisfaction instead of expected utility theory. Nevertheless, we agree that we may have overstating satisfaction as a measure of decision quality. Additionally, we agree that the measurement could be conflated with decision-maker’s internal states (e.g., anxiety, depression). However, we believe that a decision maker would be less satisfied with an anxious/depressive avoidance than with a confident avoidance. Therefore, the conflation could also be part of adaptivity. 

We now have adapted the passage by better explaining the choice of decision satisfaction as measurement of decision quality and weaken the classification by using “one suitable” instead of “well-established” measurement, see p. 13: 

“Thus, normative measurements to assess the quality of decision-making behavior (like the multi-attribute utility theory (62)) are not applicable in this context as the outputs are uncertain. Subjective measurements like the experienced utility (63) or emotions like regret (64) can be considered as alternative indicators for the quality of rational decision-making under uncertainty. Against this background, the degree of satisfaction with decision-making behavior is one suitable subjective measurement (65, 66). Inspired by the concept of choice-process satisfaction, it captures decision-making as well as decision avoidance (67). The measurement provides a subjective evaluation, so that predefined probabilities and outcomes are not necessary.” (p. 13, line 305-313)

4. Because odds ratios are expressed in the units of the predictor, does the Jeffreys’ criteria only apply when predictors are dichotomous?

-> Thank you for your interest. In the literature, the Jeffreys’ criteria are commonly used, when predictors are dichotomous, because the classifications rely on odds ratios. However, we are no experts to say for sure if this could be used for continuous predictors as well. 

5. The “time too fast (less than five times the median)” is not clear. Can the authors rephrase?

-> Thank you for your question. Every participant had a processing time for filling out the study. The online presentation program “SoSciSurvey” suggests to exclude participants that rush too fast through the questionnaire (too fast is defined as faster than five times the median of the overall processing time), because no thorough reading can be expected. We tried to clarify the term by adding more explanation, see p. 21:

“They indicated a low seriousness (< 50% on a scale 0-100) or the study was completed in such a short time (faster than than five times the median of the processing time of all study participants) so that a detailed engagement with the material was questioned.” (p.21, line 483-485)

6. There is some awkwardness in phrasing that is apparent at many points, but it is not distracting.

-> We worked through the manuscript again and adjusted the phrasing in a few places.

7. The manuscript is at times unnecessarily detailed and repetitive; a considerable reduction would improve readability.

-> We agree that the manuscript is at times detailed, partly as a result of additional wishes and explanations throughout the review process. In the end it is difficult to make significant cuts and we prefer to repeat instead of missing important aspects of the study. We hope the readability has improved.

---

## [Editor Report · Decision Letter 3]

12 Oct 2020

A vignette study of option refusal and decision deferral as two forms of decision avoidance: Situational and personal predictors

PONE-D-19-17779R3

Dear Dr. Funke,

We’re pleased to inform you that your manuscript has been judged scientifically suitable for publication and will be formally accepted for publication once it meets all outstanding technical requirements.

Kind regards,

Valerio Capraro

Academic Editor

PLOS ONE
---

## [Editor Report · Acceptance letter]

15 Oct 2020

PONE-D-19-17779R3 

A Vignette Study of Option Refusal and Decision Deferral as two
Forms of Decision Avoidance: Situational and Personal Predictors 

Dear Dr. Funke:

I'm pleased to inform you that your manuscript has been deemed suitable for publication in PLOS ONE. Congratulations! Your manuscript is now with our production department. 

Kind regards, 

on behalf of

Dr. Valerio Capraro 

Academic Editor

PLOS ONE